# Revisiting the contribution of land transport and shipping emissions to tropospheric ozone

Mariano Mertens[1], Volker Grewe[1a], Vanessa S. Rieger[1a], and Patrick Jöckel[1]

[1]Deutsches Zentrum für Luft- und Raumfahrt, Institut für Physik der Atmosphäre, Oberpfaffenhofen, Germany
[a]also at: Delft University of Technology, Aerospace Engineering, Section Aircraft Noise and Climate Effects, Delft, the Netherlands

*Correspondence to:* Mariano Mertens (mariano.mertens@dlr.de)

**Abstract.** We quantify the contribution of land transport and shipping emissions to tropospheric ozone for the first time with a chemistry-climate model including an advanced tagging method (also known as source apportionment), which considers not only the emissions of nitrogen oxides ($NO_x$, $NO$ and $NO_2$), carbon monoxide $CO$ or volatile organic compounds (VOC) separately, but also their non-linear interaction in producing ozone. For summer conditions a contribution of land transport emissions to ground level ozone of up to 18 % in North America and South Europe is estimated, which corresponds to 12 $nmol\,mol^{-1}$ and 10 $nmol\,mol^{-1}$, respectively. The simulation results indicate a contribution of shipping emissions to ground level ozone during summer in the order of up to 30 % in the Northern Pacific Ocean (up to 12 $nmol\,mol^{-1}$) and 20 % in the Northern Atlantic Ocean (12 $nmol\,mol^{-1}$). With respect to the contribution to the tropospheric ozone burden, we quantified values of 8 % and 6 % for the land transport and shipping emissions, respectively. Overall, the emissions from land transport contribute around 20 % to the net ozone production near the source regions, while shipping emissions contribute up to 52 % to the net ozone production in the Northern Pacific Ocean. To put these estimates in the context of literature values, we review previous studies. Most of them used the perturbation approach, in which the results for two simulations, one with all emissions and one with changed emissions of the source of interest, are compared. For a better comparability with these studies, we also performed additional perturbation simulations, which allow a consistent comparision of results using the perturbation and the tagging approach. The comparison shows that the results strongly depend on the chosen methodology (tagging or perturbation approach) and on the strength of the perturbation. A more in-depth analysis for the land transport emissions reveals that the two approaches give different results particularly in regions with large emissions (up to a factor of four for Europe). Our estimates of the ozone radiative forcing due to emissions of land transport and shipping emissions are, based on the tagging method, 92 $mW\,m^{-2}$ and 62 $mW\,m^{-2}$, respectively. Compared to our best estimates, previously reported values using the perturbation approach are almost a factor of 2 lower, while previous estimates using a $NO_x$ only tagging are almost a factor of 2 larger. Overall our results highlight the importance of differentiating between the perturbation and the tagging approach, as they answer two different questions. In line with previous studies, we argue that only the tagging approach (or source apportionment approaches in general) can estimate the contribution of emissions, which is important to attribute emission sources to climate change and/or extreme ozone events. The perturbation approach, however, is important to investigate the effect of an emission change. To effectively asses mitigation options both approaches should be combined. This

combination allows to track changes in the ozone production efficiency of emissions from sources which are not mitigated and shows how the ozone share caused by these unmitigated emission sources subsequently increases.

## 1 Introduction

Ozone in the troposphere has several well known effects: it contributes to global warming due to its radiative properties (e.g. Stevenson et al., 2006; Myhre et al., 2013), and large concentrations of ozone are harmful to humans and to plants (e.g. World Health Organization, 2003; Fowler et al., 2009). In addition, ozone is an important source for the OH radical, which controls the cleansing capacity of the troposphere (e.g. the lifetime of methane, Naik et al., 2013). Due to these different effects ozone is a central species of atmospheric chemistry (Monks et al., 2015).

Two important sources of ozone exist in the troposphere – the downward transport from the stratosphere and the in-situ production from precursor emissions (e.g. Lelieveld and Dentener, 2000; Grewe, 2004). The most important precursors of ozone are carbon monoxide (CO), methane ($CH_4$), volatile organic compounds (VOC) and nitrogen oxides ($NO_x$=NO+$NO_2$, e.g. Haagen-Smit, 1952; Crutzen, 1974; Monks, 2005). These precursors have anthropogenic as well as natural sources. Important natural sources of VOCs are biogenic emissions (e.g. Guenther et al., 1995), while $NO_x$ is emitted by lightning (e.g. Schumann and Huntrieser, 2007) and soil (e.g. Yienger and Levy, 1995; Vinken et al., 2014). Anthropogenic sources of ozone precursors, on the other hand, include emissions from industry, land transport (containing the sources road traffic, inland navigation and railways, e.g. Uherek et al., 2010) and shipping (e.g. Eyring et al., 2010). With respect to the influence of different emission sources on ozone itself, typically two different questions are of interest (e.g. Wang et al., 2009; Grewe et al., 2010; Clappier et al., 2017):

– How sensitive does ozone respond to changes of a specific emission source (sensitivity study)?

– How large is the contribution of different emission sources to ozone (source apportionment)?

Sensitivity studies are important to investigate the influence of an emission change on, for instance, ozone. Often, the so called perturbation approach has been applied, in which the results of two (or more) simulations are compared: one reference simulation with all emissions and a sensitivity simulation with perturbed emissions. Source apportionment, in contrast, is important to attribute different emission sources to climate impact (such as radiative forcing) or extreme ozone events. Source apportionment studies often use tagged tracers in order to estimate contributions of different emission sources, for instance, to ozone. In this tagging approach, additional diagnostic species are introduced which follow the reaction pathways of the emissions from different sources (e.g. Lelieveld and Dentener, 2000; Dunker et al., 2002; Grewe, 2004; Gromov et al., 2010; Butler et al., 2011; Grewe et al., 2012; Emmons et al., 2012; Kwok et al., 2015). Other methods exist for both type of studies (e.g. sensitivity and source apportionment), which we neglect here for simplicity (see e.g. Clappier et al., 2017).

In a linear system, both approaches, perturbation and tagging, lead to the same result (e.g. Grewe et al., 2010; Clappier et al., 2017). The $O_3$ chemistry, however, is highly non-linear. Therefore, both approaches lead to different results, not because of uncertainties in the method, but because they give answers to different questions. Here, we use the following wording to

discriminate between these two types of questions and methods, knowing that other authors may use them differently: The impact of a source is calculated by the sensitivity method (here the perturbation approach), while the contribution is calculated using a source apportionment method (here tagging approach, e.g., Wang et al., 2009; Grewe et al., 2010; Clappier et al., 2017). Accordingly, the impact indicates the effect of an emissions change, while the contribution enables an attribution of ozone (and associated radiative forcing) to specific emissions sources.

In the past, many studies have been performed to estimate the impact of road traffic emissions (but not the total land transport effect, e.g., Granier and Brasseur, 2003; Niemeier et al., 2006; Matthes et al., 2007; Hoor et al., 2009; Koffi et al., 2010) on the global scale. However, only few studies exist estimating the contribution of road traffic emissions on ozone: Dahlmann et al. (2011) and Grewe et al. (2012) used a tagging approach considering only $NO_x$. Further, these studies focussed mainly on globally averaged tropospheric ozone columns and associated radiative forcings without regional quantifications of the contribution. Similar, for the shipping sector previous studies focused on the calculation of the impact (e.g. Lawrence and Crutzen, 1999; Eyring et al., 2007; Hoor et al., 2009; Koffi et al., 2010; Holmes et al., 2014). Only Dahlmann et al. (2011) reported results of the $O_3$ due to shipping emissions using a $NO_x$-only tagging approach.

It is well known that the impact is usually smaller compared to the contribution (e.g. Grewe et al., 2012; Emmons et al., 2012; Grewe et al., 2017). Furthermore, impacts are usually not additive. This means that the ozone changes (impacts) which are calculated for different emission sources by perturbing one of the emission source is not the same as perturbing all of the emission sources at the same time. This does not only hold for the ozone concentration but also for the associated ozone radiative forcing. As land traffic and shipping emissions are important sources of ozone precursors, it is very important to calculate not only their impact on ozone, but also the contribution of these emissions to ozone in detail. Further, our approach tags for the first time not only $NO_x$ and VOC individually, but both ozone precursors concurrently (Grewe et al., 2017). Therefore, the goal of the present study is twofold: first we review estimates of the contribution and impact of land transport and shipping emissions on tropospheric ozone and the resulting radiative forcing. Second, we present new results analysing the contribution of land transport and shipping emissions in detail using a tagging approach. These new results quantify for the first time the contributions of the considered emissions on (ground-level) ozone in detail. Further, we also report results using a perturbation approach in a consistent manner to bridge the gap between previous studies and our new results. This allows a detailed comparison of the impact and contribution, as well as the associated ozone radiative forcings of land transport and shipping emissions between the perturbation approach, $NO_x$-tagging and $NO_x$-&VOC tagging.

The paper is organised as follows: In Sect. 2 we give an overview of the used model system and describe the applied set-up. In Sect. 3 we analyse our simulation results with respect to contribution versus impact of land transport and shipping emissions to ground level ozone including a detailed overview and discussion of the results from previous studies. In Sect. 4 we compare our results using the perturbation and the tagging approach in more detail. Section 5 gives more detailed insights into the tropospheric ozone budget. The contribution of the land transport and shipping emissions to radiative forcing due to ozone is analysed in Sect. 6, while Sect. 7 gives a discussion about the uncertainties associated with the tagging and the perturbation approaches, respectively.

## 2 Model description and set-up

### 2.1 Model description

We applied the ECHAM/MESSy Atmospheric Chemistry (EMAC) chemistry-climate model (Jöckel et al., 2006, 2010, 2016) equipped with the TAGGING technique described by Grewe et al. (2017). EMAC uses the second version of the Modular Earth Submodel System (MESSy2) to link multi-institutional computer codes. The core atmospheric model is the 5th generation European Centre Hamburg general circulation model (ECHAM5 Roeckner et al., 2006). For the present study we applied EMAC (ECHAM5 version 5.3.02, MESSy version 2.52) in the T42L90MA-resolution, i.e. with a spherical truncation of T42 (corresponding to a quadratic Gaussian grid of approx. 2.8 by 2.8 degrees in latitude and longitude) with 90 vertical hybrid pressure levels up to 0.01 hPa. The simulation set-up is almost identical to the one of the simulation *RC1SD-base-10a* described in detail by Jöckel et al. (2016) alongside with an evaluation of the resulting model simulation. Therefore, we describe only the most important details and differences. A comparison with the results of the simulation presented here and the *RC1SD-base-10a* is part of the Supplement of the present manuscript.

The chosen simulation period covers the years 2004 to 2010. The years 2004–2005 serve as spin-up, while the years 2006–2010 are analysed. Initial conditions for the trace gas distribution were taken from the *RC1SD-base-10a* simulation (Jöckel et al., 2016). Lightning $NO_x$ is parameterised after Grewe et al. (2002) with global total emissions of $\approx 4.5$ Tg(N) $a^{-1}$. Emissions of $NO_x$ from soil and biogenic $C_5H_8$ emissions were calculated using the MESSy submodel ONEMIS (Kerkweg et al., 2006), using parameterisations based on Yienger and Levy (1995) for soil-$NO_x$ and Guenther et al. (1995) for biogenic $C_5H_8$. The applied gas phase mechanism in MECCA (Sander et al., 2011) incorporates the chemistry of ozone, methane and odd nitrogen. Alkanes and alkenes are considered up to C4, while the oxidation of $C_5H_8$ and some non-methane hydrocarbons (NMHCs) are described with the Mainz Isopren Mechanism version 1 (von Kuhlmann et al., 2004). Further, heterogeneous reactions in the stratosphere (submodel MSBM, Jöckel et al., 2010) as well as aqueous phase chemistry and scavanging (SCAV, Tost et al., 2006) are included. Emissions of methane ($CH_4$) are not considered explicitly. Instead pseudo-emissions are calculated using the submodel TNUDGE (Kerkweg et al., 2006). TNUDGE relaxes mixing ratios in the lowest model layer towards observations using Newtonian relaxation (see also Jöckel et al., 2016).

EMAC is 'nudged' by Newtonian relaxation of temperature, divergence, vorticity and the logarithm of surface pressure (Jöckel et al., 2006) towards ERA-Interim (Dee et al., 2011) reanalysis data. Also the sea surface temperature and sea ice coverage are prescribed as transient time-series from ERA-Interim too. To allow for identical meteorological conditions in sensitivity experiments with changed emissions, the quasi chemistry transport model mode (QCTM-mode, Deckert et al., 2011) of EMAC was used. In this mode, climatologies of the radiative active trace gases are prescribed for the calculation of the radiation. Further, climatologies are used for processes which couple the chemistry and the hydrological cycle. The applied climatologies are monthly average values taken from the *RC1SD-base-10a* simulation.

## 2.2 Tagging method for source attribution

The tagging is performed using the MESSy TAGGING submodel described in detail by Grewe et al. (2017). This tagging method is an accounting system following the relevant reaction pathways and applies the generalized tagging method introduced by Grewe (2013). This method diagnoses the contributions of different categories to the regarded species without influencing the full chemistry. A prerequisite for this method is a complete decomposition of the source terms, e.g. emissions, of the regarded species in $N$ unique categories. As a consequence of the complete decomposition, the sum of the contributions of all tagged categories of one specie equals the total concentration of this specie (i.e. the budget is closed):

$$\sum_{\mathrm{tag}=1}^{N} \mathrm{O}_3^{\mathrm{tag}} \quad = \quad \mathrm{O}_3. \tag{1}$$

As an example of this method consider the production of $\mathrm{O}_3$ by the reaction of NO with an organic peroxy radical ($\mathrm{RO}_2$) to $\mathrm{NO}_2$ and the organic oxy radical (RO):

$$\mathrm{NO} + \mathrm{RO}_2 \longrightarrow \mathrm{NO}_2 + \mathrm{RO}. \tag{R1}$$

For this reaction the tagging approach leads to the following fractional apportionment (c.f. Eq. 13 and 14 in Grewe et al., 2017, for a detailed example):

$$\mathrm{P}_{\mathrm{R1}}^{\mathrm{tag}} \quad = \tfrac{1}{2}\mathrm{P}_{\mathrm{R1}} \left( \frac{\mathrm{NO}_{\mathrm{y}}^{\mathrm{tag}}}{\mathrm{NO}_{\mathrm{y}}} + \frac{\mathrm{NMHC}^{\mathrm{tag}}}{\mathrm{NMHC}} \right). \tag{2}$$

In this case the variables marked with $^{\mathrm{tag}}$ represent the tagged production rate of $\mathrm{O}_3$ by reaction R1 ($\mathrm{P}_{\mathrm{R1}}$) as well as the tagged families of $\mathrm{NO}_{\mathrm{y}}$ and NMHC (details given below) of one individual category (e.g. land transport). Accordingly the fractional apportionment is inherent to the method based on a combinatorial approach, which decomposes every regarded reaction into all possible combinations of reacting tagged species. This takes into account the specific reaction rate constant from the full chemistry scheme (implicitly by the production and loss rates from the chemistry solver). The chemical mechanism including all diagnosed production and loss rates for the tagging method are part of the Supplement. The analysed production and loss rates in Sect. 5 are calculated in accordance with Eq. 13 and 14 of Grewe et al. (2017).

The applied method considers ten categories (detailed definition is given in Table 1). To minimize the needed amount of memory and computational performance, not every individual specie is tagged. Instead a family concept is chosen. The following families are taking into account: $\mathrm{O}_3$, $\mathrm{NO}_{\mathrm{y}}$, PAN, NMHC and CO. Additionally, OH and $\mathrm{HO}_2$ are tagged by a steady state approach. In the following, we denote absolute contributions of land transport and shipping emissions to ozone diagnosed with the tagging method as $\mathrm{O}_3^{\mathrm{tra}}$ and $\mathrm{O}_3^{\mathrm{shp}}$, respectively.

## 2.3 Radiative forcing

The radiative forcing (RF) of ozone is defined as the difference of the net radiative fluxes caused by a change (e.g. between two time periods like pre-industrial and present day, Myhre et al., 2013). Here, we are interested in the contribution of land transport and shipping to this RF. Due to the non-linearities in the ozone chemistry (see also Sect. 4), we estimate the contribution of the land transport/shipping emissions to ozone and then calculate the RF of these $O_3$ shares individually. This approach is consistent with the IPCC RF definition, since the sum of all individual RF contributions approximately equals the total RF (for a detailed example see Dahlmann et al., 2011).

Thus, to calculate the $O_3$ RFs of land traffic and shipping emissions, additional simulations were performed applying the stratospheric adjusted radiative forcing concept (e.g. Hansen et al., 1997; Stuber et al., 2001; Dietmüller et al., 2016). For this, monthly mean fields of the simulation *RC1SD-base-10a* are used as input data, of the radiation scheme, except for $O_3$, which stem from the *BASE* simulation. Calculations of the RF based on the results of the tagging approach in accordance with Dahlmann et al. (2011) were performed as follows:

1. Based on the results of the *BASE* simulation, monthly mean values of $\Delta_T^{\mathrm{tra}} = O_3 - O_3^{\mathrm{tra}}$ and $\Delta_T^{\mathrm{shp}} = O_3 - O_3^{\mathrm{shp}}$ were calculated. $\Delta_T^{\mathrm{tra}}$ and $\Delta_T^{\mathrm{shp}}$ corresponds to the share of $O_3$ excluding $O_3$ from land transport and shipping emissions, respectively.

2. Multiple radiation calculations (Dietmüller et al., 2016) were performed, calculating the radiative flux of $\Delta_T^{\mathrm{tra}}$, $\Delta_T^{\mathrm{shp}}$ and $O_3$. The $O_3$ RFs of land transport and shipping emissions using the tagging approach are then calculated as follows:

$$\mathrm{RF}_{\mathrm{O3tra}}^{\mathrm{tagging}} = rflux(O_3) - rflux(\Delta_T^{\mathrm{tra}}), \tag{3}$$

$$\mathrm{RF}_{\mathrm{O3shp}}^{\mathrm{tagging}} = rflux(O_3) - rflux(\Delta_T^{\mathrm{shp}}), \tag{4}$$

with $rflux$ being the net radiative fluxes calculated for the respective quantity. Accordingly, the calculated RFs measure the flux change caused by the ozone share of land transport and shipping emissions, respectively.

Calculating the RFs based on the results of the perturbation approach is similar to (e.g. Myhre et al., 2011). First, $\Delta O_{3\mathrm{tra}}$ and $\Delta O_{3\mathrm{shp}}$ are calculated by taking the difference between the unperturbed (*BASE*, see below) and the perturbed simulations (*LTRA95* or *SHIP95*):

$$\Delta O_3 = (O_3^{\mathrm{unperturbed}} - O_3^{\mathrm{perturbed}}) \cdot 20. \tag{5}$$

As we consider 5 % perturbations (e.g. the emissions of land transport and shipping are decreased by 5 %, see Sect. 2.4) these differences are scaled by a factor of 20 to yield a 100 % perturbation. To calculate the RFs using the perturbation approach, $\Delta O_{3\mathrm{tra}}$ and $\Delta O_{3\mathrm{shp}}$ are than treated as described above for $\Delta_T^{\mathrm{tra}}$ and $\Delta_T^{\mathrm{shp}}$. These RFs are called $\mathrm{RF}_{\Delta\mathrm{O3tra}}^{\mathrm{perturbation}}$

and $\text{RF}^{\text{perturbation}}_{\Delta\text{O3shp}}$, respectively. Accordingly, the method to calculate the RFs of the $O_3$ shares analysed by the perturbation and the tagging approach are the same. The differences between $\text{RF}^{\text{perturbation}}_{\text{O3tra}}$ and $\text{RF}^{\text{tagging}}_{\text{O3tra}}$ (and the same for shipping) arise only due to differences of the the differently calculated $O_3$ shares.

The benefit of using the contribution of an emission source (in contrast to using the impact of the emission source) is that for

the contribution the sum of the individual radiative forcings is equal to the total RF, i.e. $\sum_i^n RF^i \approx RF$ with $RF^i$ being the radiative forcings of the individual categories $i$ of $n$ total categories. This hold for the perturbation approach (Dahlmann et al., 2011; Grewe et al., 2012). However, the calculations of the RF is still subject to some specific assumptions, which we discuss in detail in the Supplement.

In general, we consider only the direct RF due to changes of the $O_3$ concentration. We calculate no RF due to changes of

the methane concentration caused by anthropogenic emissions. These changes would lead to a negative RF due to decreased methane concentrations. Especially for shipping emissions the negative RF due to methane can be larger compared to the positive ozone forcing (e.g. Myhre et al., 2011).

## 2.4   Simulation set-up

As anthropogenic emissions inventory we chose the MACCity emission inventory (Granier et al., 2011), which follows the

RCP8.5 scenario (Riahi et al., 2007, 2011) for the analysed period. The monthly varying anthropogenic emissions are represented on a grid with $0.5°$ x $0.5°$ spatial resolution. The geographical distribution of the land transport (containing road traffic, inland navigation and railways) and the shipping sector are shown in Fig. 1. Additionally, the total emissions of CO, $NO_x$ and NMHCs of the most important emission sectors are given in Table 2.

Three different simulations were conducted: one with all emissions (*BASE*), one with a 5 % decrease of the land transport

emissions of $NO_x$, CO and VOCs (*LTRA95*), and one with a 5 % decrease of the shipping emissions of $NO_x$, CO and VOCs (*SHIP95*). The 5 % perturbation was chosen as previous studies showed that this small perturbation sufficiently minimises the impact of the non-linearity of the chemistry on the results (e.g. Hoor et al., 2009; Grewe et al., 2010; Koffi et al., 2010).

All three simulations were equipped with the full tagging diagnostics. To quantify the contribution of the emission sources the tagging results of the *BASE* simulation are used. The simulations with a decrease of the land transport and shipping

emissions were performed to allow for a direct comparison between the tagging and the perturbation method. The additional tagging diagnostics in the perturbed simulations allow for a more detailed investigations in the change of the ozone production (see Sect. 4).

In the present study we focus on the source regions of land transport and shipping emissions. Therefore we use the same geographical regions as defined by Righi et al. (2013) to investigate the contribution these emissions. The regions are Europe

(EU), North America (NA) and Southeast Asia (SEA) for land transport, and North Atlantic Ocean (NAO), Indian Ocean (IO) and North Pacific Ocean (NPO) for the shipping emissions.

## 3   Contribution to ground level ozone

First, we analyse the absolute amount of $O_3$ produced by land transport (tra) and ship (shp) exhaust as analysed with the tagging approach. Additionally we indicate also the relative contribution of $O_3^{tra}$ and $O_3^{shp}$ to near ground level $O_3$. For all quantities multi-annual, seasonal average values for December–February (DJF) as well as June–August (JJA) for the years 2006–2010 (for DJF starting with December 2005) were computed.

### 3.1   Land transport

Figure 2a and Fig. 2b show the seasonal average values of $O_3^{tra}$ for DJF and JJA . The maximum absolute contribution for each hemisphere are simulated during local summer conditions when the photochemistry is most effective. Most geographical locations of these maxima correspond to the regions with the largest land transport emissions. The largest absolute contributions of 8–14 $nmol\ mol^{-1}$ are simulated during JJA on the Northern Hemisphere in North America (8–12 $nmol\ mol^{-1}$), Southern Europe (8–10 $nmol\ mol^{-1}$), the Arabian Pensinsula (12–14 $nmol\ mol^{-1}$), India (8–10 $nmol\ mol^{-1}$) and Southeast Asia (6–10 $nmol\ mol^{-1}$). In Asia the largest values are simulated around the Korean Peninsula rather than in China. This lower contribution of land transport emissions in China compared to Europe or North America is mainly caused by a much larger fraction of other anthropogenic emissions (e.g. industry and households) compared to land transport emissions (e.g. Righi et al., 2013). Accordingly much more $O_3$ is produced in China by other anthropogenic emissions compared to land transport. The local maxima (4–6 $nmol\ mol^{-1}$) on the Southern Hemisphere are simulated during DJF, when the photochemistry is most active. These maxima are located in South America and South Africa. Corresponding the regions with the largest land transport emissions on the Southern hemisphere (cf. Fig. 1).

The relative contribution of $O_3^{tra}$ to near ground level $O_3$ is depicted in Fig. 2c and Fig. 2d. Values of 14–16 % are simulated during DJF around the source regions on the Southern Hemisphere, but the absolute values on the Southern Hemisphere are lower compared to the Northern Hemisphere. The simulated relative contributions on the Northern Hemisphere during DJF is around 10 %. Only around the Arabian Peninsula values of 14–16 % are found. During JJA, these maxima increase to 14–18 % over North America and 12–16 % for the other hotspot regions on the Northern Hemisphere. One important reason for the change of the contribution from DJF to JJA (on the Northern Hemisphere) is the strong seasonal cycle of the anthropogenic non-traffic sector in our applied emission inventory, showing large emissions during winter and lower emissions during summer. This leads to larger contributions of the anthropogenic non-traffic category during DJF compared to JJA.

To review estimates of the impact and contribution of previous studies and to compare the new results with previous values, Table 3 summarises the amount of emissions as well as reported impacts/contributions of road traffic emissions from previous studies. So far, only the effects of road traffic emissions alone and not the total effect of land transport emissions have been investigated. With respect to the ozone precursors road traffic emissions are the largest contributor to the land transport sector. The contributions of inland navigation and railways are smaller than the uncertainties of the road traffic emissions. Therefore we argue that our results of the land transport sector can be compared with previous studies considering only road traffic emissions (cf. also the amount of applied emissions in different studies in Table 3). In general, we are focussing on global studies only.

Regional effects of road traffic emissions have been investigated too (e.g. Reis et al., 2000; Tagaris et al., 2015; Hendricks et al., 2017), but because of the coarse resolution of global models a quantitative comparison between findings of regional studies with these global studies is not straightforward and probably not meaningful. Please note that we list our values in Table 3 for July conditions only, to be comparable to other studies, since they also reported values for July conditions. In addition the impact of the land transport emissions were calculated by with the results of the unperturbed and perturbed simulation (*BASE* minus *LTRA95*) which is scaled by 20 to estimate a 100 % perturbation. Figures showing the contribution/impact for the results of the present study are part of the Supplement.

Previously, the impact of road traffic emissions on ozone concentration has been investigated mainly using 100 % and 5 % perturbation approaches. Most previous studies applied similar amounts of road traffic emissions as the present study used for land transport emissions (9–10 $\text{Tg a}^{-1}$). The fraction of $NO_x$ emissions from road traffic compared to all emissions was largest in the studies of Granier and Brasseur (2003), Niemeier et al. (2006) and Matthes et al. (2007). These studies also applied the largest CO and VOC emissions, while the individual fractions vary across the studies.

In general, the results of all considered studies can be separated into three groups: (1) The largest values are reported by the present study (using the tagging approach) as well as by Niemeier et al. (2006). (2) Slightly lower values are given by Granier and Brasseur (2003) and Matthes et al. (2007), while (3) Hoor et al. (2009) and Koffi et al. (2010) report the lowest impact. These studies, however, differ not only in the emission inventories and models used, but also in the methods. The lowest values are in general reported by studies using the 5 % perturbation (scaled to 100 %), which is confirmed by our results using the same method. However, in general our simulation results show larger values compared to these previous findings. These differences are noticeable especially for the NA region. The differences might be caused by a different geographical distribution of the emissions, as well by larger CO and NMHC emissions in the emission inventory we applied. Further, differences in the atmospheric composition as simulated by the different models can influence the production rates of ozone, which might contribute to the differences of the simulated impacts.

The comparison of our results using the 5 % perturbation approach with the results using the tagging approach clearly confirms the known differences between estimates of the impact (perturbation) and contribution (tagging, e.g. Wang et al., 2009; Grewe et al., 2010; Emmons et al., 2012; Grewe et al., 2012, 2017; Clappier et al., 2017). Depending on the region, we find a difference of up to a factor of 4. The reason for this difference is investigated in more detail in Sect. 4.

Granier and Brasseur (2003), Niemeier et al. (2006) and Matthes et al. (2007), however, also used a perturbation approach, but report values, which are more similar to our estimate using the tagging method. This is likely caused by the larger emissions applied in these studies compared to all other studies. Accordingly, the contribution of the road traffic emissions is underestimated by the perturbation method, but the larger emissions (and fraction) of the road traffic category lead to results, which are similar as estimated by the tagging method with smaller emissions. Of course also other factors, like differences between the models, chemical mechanisms, geographical distribution, and different seasonal cycles of the emissions can contribute to differences between the studies. The influence of these factors, however, is difficult to reveal.

## 3.2 Ship traffic

The absolute contribution of $O_3^{shp}$ are shown in Fig. 3a and Fig. 3b. Similar to the shipping emissions (cf. Fig. 1), $O_3^{shp}$ shows a strong North-South gradient. The maximum values in the Northern Hemisphere are located between $20°$–$30°$ N during DJF ($\approx 6\ \mathrm{nmol\ mol^{-1}}$). These maxima move northwards during summer and increase in magnitude ($10$–$12\ \mathrm{nmol\ mol^{-1}}$). This shift is caused by the increase in the photochemical activity in the Northern hemisphere during summer. Most shipping emissions are located north of $30°$ N (see Fig. 1). With increasing ozone production during spring and summer more $O_3^{shp}$ near the regions with the largest emissions are formed, compared to the regions of $20$–$30°$ N.

The largest values of the relative contribution of $O_3^{shp}$ during DJF are around 14 % and are co-located with the regions of the largest values of $O_3^{shp}$ (Fig. 3c). The maxima of the contribution increase during JJA to around 30 % in the Northwestern Pacific, while the values in the Northeastern Pacific are around 18–22 %. In the Northern Atlantic maximum contributions of 20 % are simulated (Fig. 3d).

Table 4 summarises emissions and results of previous studies. In general most studies used similar global $NO_x$ shipping emissions of around $4\ \mathrm{Tg(N)\ a^{-1}}$. The largest impact/contribution of shipping emissions is limited to distinct areas within the investigated geographical regions. Therefore the range of the given contributions/impacts within the geographical regions is large. The displacement between the regions of emissions and largest ozone production is well known (e.g. Endresen et al., 2003; Eyring et al., 2007) and mainly caused by complex interplay between $NO_x$ emissions, transport of precursors and ozone production.

Similar as discussed for the impact/contribution of land transport emissions, there is a large discrepancy between the results using the 100 % and the 5 % perturbation method. The studies using the 100 % method report impacts in the Atlantic and the Pacific in the range of $4$–$11\ \mathrm{nmol\ mol^{-1}}$ (corresponding to 12–40 %). In general the previous studies report larger impacts in the Pacific compared to the Atlantic. Only Eyring et al. (2007) reported a larger perturbation in the Northern Atlantic compared to the Pacific, which can most likely be attributed to differences in the emission inventories, as Eyring et al. (2007) applied lower emissions in the Northern Pacific compared to the Northern Atlantic.

Hoor et al. (2009) and Koffi et al. (2010) report absolute impacts (5 % perturbation) in the range of $2$–$6\ \mathrm{nmol\ mol^{-1}}$. Our model results using a 5 % perturbation suggest somewhat larger impacts of around $2$–$8\ \mathrm{nmol\ mol^{-1}}$ (10–22 %) in the Atlantic and Pacific. Most likely this difference can be attributed to different shipping emissions applied.

The absolute contributions diagnosed using the tagging approach are larger and in the range of $3$–$11\ \mathrm{nmol\ mol^{-1}}$ (relative contribution: 10–33 %) in the Atlantic and Pacific. These contributions are at the lower end of the contributions reported by the studies using the 100 % approach. Compared to these studies, however, we applied the largest shipping emissions. Accordingly, a larger contribution compared to other studies can be expected. As the used models and emission inventories in all studies are very different we can only speculate about possible reasons.

One reason for this discrepancy might be the resolution of the model simulations. In previous studies a variety of resolutions were used (especially in the multi model approaches by Eyring et al. (2007) and Hoor et al. (2009). Our horizontal resolution of $\approx 2.8°$ is at the finer end of most of these resolutions (only Dalsøren et al. (2009) used $\approx 1.875°$). A coarse resolution leads

to a strong dilution of the shipping emissions. This effect can lead to an overestimation of the $O_3$ production (e.g. Wild and Prather, 2006). Our results are also influenced by this problem too, because a resolution of T42 dilutes the emissions over large areas. A model with finer resolution, effective emissions, or a plume model (e.g. Franke et al., 2008; Holmes et al., 2014) diagnoses likely smaller contributions. Another important contributor to the differences is the geographical distribution of ship emissions. If the ship tracks are too narrow, the ozone production might be suppressed (see discussion by Eyring et al., 2007). Further, differences in the seasonal cycles of emissions con contribute to the differences.

## 4    Comparing perturbation and tagging approach

As discussed in the previous section and by previous studies (e.g. Wang et al., 2009; Grewe et al., 2010) the perturbation approach, which is often used for source attribution, and the tagging approach lead to different results. To investigate this effect in more detail, $\Delta O_{3\mathrm{tra}}$ (see Eq. 5) is analysed further. Here, we consider not only ground-level values, but partial ozone columns integrated from the surface up to 850 hPa (called 850PC, in DU).

To quantify the difference between the perturbation and the tagging approach in more detail, Fig. 4a shows the 850PC of $\Delta O_{3\mathrm{tra}}$. Figure 4b shows the 850PC of ($O_3^{\mathrm{tra}}$) for the *BASE* simulation. A qualitative comparison already indicates a relative large difference between the impact (as estimated by the perturbation approach, Fig. 4a) and the contribution (by the tagging approach, Fig. 4b). Figure 4c shows the relative difference between both quantities, indicating a difference between 40–80 %. The lowest differences are found on the Southern Hemisphere, while the difference is largest near the hotspot regions (North America, Europe and South-East Asia). Here, the impact is up to a factor of four lower compared to the contribution (not shown). A large relative difference is also indicated in some regions near the equator. In these regions, however, the absolute difference is low. The only region where a difference below 20 % is simulated is in parts of South America. This difference between the impact and the contribution is not confined to the lower troposphere, but is present throughout the troposphere (additional figures showing zonal averaged impact and contributions are part of the Supplement).

To further investigate why the difference between impact and contribution largely change between the regions, the dependency between $NO_x$ mixing ratios (caused by changes of the emissions) and the net $O_3$ production of the results for the year 2010 is analysed. Figure. 5 shows this dependency for the whole globe (black) and some chosen areas (coloured dots). Generally the well known dependency (e.g. Seinfeld and Pandis, 2006) between $O_3$ production and $NO_x$ concentrations can be observed. In pristine regions a net loss of $O_3$ is, present (first regime). With increasing $NO_x$ mixing ratios the net $O_3$ production increases strongly (called $NO_x$-limited regime). The production of $O_3$ decreases again with even larger $NO_x$ values. In this third regime, however, the production of $O_3$ can be increased if the NMHC emissions are increased (called NMHC-limited regime). Every dot represents a different grid box of the model with different meteorological conditions and background mixing ratios of CO, NMHC etc. Therefore, the dependency between the $NO_x$ mixing ratio and the net $O_3$ production differs for every grid box and is not given by one single function (which is the case for boxmodel calculations with prescribed conditions).

In different regions of the world the $O_3$ production takes place in different chemical regimes, depending on the amount of $NO_x$ emissions. Therefore, the coloured dots highlight the individual relationship between $NO_x$ mixing ratio and production

of $O_3$ for four different regions. Depending on the chemical regime in the different regions, the ozone chemistry responds differently to the perturbation applied in the perturbation approach (e.g., Dahlmann et al., 2011).

Based on the results of the *REF* and *LTRA95* simulations, the ozone sensitivity is calculated with the tangent approach in accordance with Grewe et al. (2010) by solving a linear equation ($y = m \cdot (x - x_0) + b$, see Supplement for additional Figures).

Here, $x$ and $y$ are the average $NO_x$ mixing ratio and the net $O_3$ production ($P_{O3}$), respectively, for a particular region. The $m$ denotes the slope, which corresponds to an approximation of the derivative $dP_{O3}/dNO_x$ in the unperturbed simulation, which is calculated by the difference in ozone production and $NO_x$ mixing ratios in the unperturbed and perturbed simulation. $x_0 = NO_x^u$ is the $NO_x$ mean mixing ratio in the unperturbed simulation and $b = P_{O3}^u - dP_{O3}/dNO_x \, NO_x^u$, where $P_{O3}^u$ is the mean ozone production in the unperturbed simulation.

Based on the linearised ozone production ($P_{O3}^{lin}$) calculated by the tangent approach, we define a saturation indicator $\Gamma$, which helps to analyse the ozone sensitivity further:

$$\Gamma = \frac{y - \text{axis intercept}}{y - \text{value of unperturbed simulation}} = \frac{P_{O3}^{lin}(NO_x = 0)}{P_{O3}^{lin}(NO_x = \text{unperturbed})}. \tag{6}$$

Accordingly, $\Gamma$ compares the production rate of ozone of the base case with unperturbed emissions ($NO_x = \text{unperturbed}$) with the approximated production rate of ozone, if $NO_x$ emissions are set to zero ($NO_x = 0$), assuming a linear ozone chemistry. This value is a quantitative indicator of the chemical regime, showing how much an emission change of one specific sector is compensated by increased ozone productivity of other sectors. $\Gamma = 1$ indicates a saturated behaviour of the ozone production i.e. the ozone production does not change, if emissions are changed ($P_{O3}^{lin}(NO_x = 0) = P_{O3}^{lin}(NO_x = \text{unperturbed})$). Accordingly, there is no ozone reduction because the change of the emissions is entirely compensated by an increasing ozone production efficiency of other emissions. $\Gamma > 1$ indicates an overcompensating effect, i.e., reduced $NO_x$ emissions lead to an increase of the ozone production (corresponding to the VOC-limited regime). Finally, $\Gamma = 0$ indicates a linear response of the system (with a y-intercept at zero). Accordingly, the ozone change introduced by an emission change is not compensated by an increase of the ozone production efficiency. For $\Gamma = 0.5$ the ozone change is half compensated by a change in the ozone production efficiency. In terms of the estimated derivative ($dP_{O3}/dNO_x$), $\Gamma = 1$ corresponds to $dP_{O3}/dNO_x = 0$, while $\Gamma > 1$ corresponds to $dP_{O3}/dNO_x < 0$ and vice versa.

Table 5 lists the $\Gamma$ values of the four different regions together with a brief interpretation of these values (additional information and figures concerning $\Gamma$ are part of the Supplement). In general, only the regions North Africa and South America show a response of the $O_3$ chemistry, which is close to linear ($\Gamma = 0.2 - 0.3$). As known (e.g. Wang et al., 2009; Grewe et al., 2010; Clappier et al., 2017) only for this linear case the perturbation and the tagging approach lead to the same results (e.g. the contribution can be estimated using a perturbation approach). In all other regions the contribution is largely underestimated by the perturbation approach.

This underlines the importance of discriminating between tagging and perturbation. Clearly, both approaches answer different, but equally important questions. The perturbation approach answers the question on the impact of an emission change. This approach is important to estimate effects due to mitigation measures (e.g. Williams et al., 2014). The tagging approach in

contrast, disentangles the ozone budget into the contributions of the individual emission sources and is important to investigate e.g. the contribution of radiative forcing of individual emission sources (see Sect. 6) or to quantify contribution of different emission sources to extreme ozone events. However, the tagging approach can not be used to quantify the impact of an emission change, while the perturbation approach should not be used to quantify the contribution. As demonstrated, in regions where

ozone responses more linearly to emission changes, both approaches differ slightly, but in regions where large emissions occur (e.g., Europe, South-East Asia) the perturbation approach largely underestimates the contributions and should not be used for source apportionment. However, if mitigation options are investigated the tagging approach should be combined with the perturbation approach (see next subsection).

## 4.1 Combining Tagging and Perturbation approach in mitigation studies

The tagging approach does not give any information about the sensitivity of the ozone chemistry with respect to a change of emissions. Accordingly, the success of an emission reduction, e.g. measured in terms of reduced ozone concentration, is evaluated using the perturbation approach. Wang et al. (2009) proposed to first use a tagging simulation estimating these sources. which contribute largest to ozone and therefore have the largest mitigation potential. However, we propose to equip all simulations, the unperturbed reference simulation and all simulations with changed emissions, with the tagging approach.

In this case the results of the perturbed simulations quantify the changes in ozone due to mitigation options. The tagging results provide additional information, which are important to quantify the accountability of different emission sources to the ozone concentration or the associated radiative forcing. These additional information are important, because the success of one specific mitigation option largely depends on the history of previous mitigations (Grewe et al., 2012).

  To present the benefits of combining both methods in more detail, Fig. 6 sketches an idalised example of four different

mitigation options. For each of the idealised mitigation options we assume a decrease of the emissions of one specific emission source by 10 arbitrary units. Mitigation option 1 reduces the land transport emissions, mitigation option 2 the shipping emissions and mitigation option 3 the emissions from industry.

  With respect to the ozone concentration (Fig. 6a) only mitigation option 3 is successful in largely reducing the ozone concentration. Having only the results with respect to the ozone concentration in mind one could attribute the ozone change

completely to the emissions change of the industry sector. From this point of view there would be no benefit to reduce land transport or shipping emissions.

  However, if all simulations are additionally equipped with a tagging method the contribution of the different emission sources to the ozone concentration is anlysed (Fig. 6b). For each of the considered cases both, the ozone concentration, and the contribution of the different emission sources to this ozone concentration differ. This additional contribution analysis shows

that even if due to mitigation option 1 the overall ozone concentration increases, the contribution of the road traffic emissions is lowered. At the same time the contribution of all other emission sources, which are not changed, increase, because the ozone production efficiency increase. However, if every emission source is made responsible for their individual contributions to ozone levels (for air quality mitigation purpose) or their individual contributions to ozone radiative forcing (for climate

mitigation purpose), an obvious benefit exists for a specific emission source to reduce it's emissions even if overall $O_3$ levels are only slightly reduced. These additional information are only available using the tagging approach.

This gets even more clear, if mitigation option 2 is considered in which the shipping emissions are reduced. The overall ozone concentration remains unchanged, as the ozone chemistry is in a saturated regime ($\Gamma = 1$). The contribution of the shipping emissions, however, decrease strongly, while the contribution of emissions from industry and household increase. Accordingly, the emission sources household and industry are more responsible for the ozone values and/or ozone radiative forcing, while the emission sources road traffic and shipping are less responsible. This puts pressure onto these emission sources to reduce emissions of ozone precursors.

In mitigation option 3 the emissions of the industry sector are reduced. In this case, the response of the ozone concentration to emission changes is close to linear ($\Gamma \approx 0$) and the ozone concentration is reduced strongly. This emission reduction causes a reduction of the ozone production efficiency, leading not only to a reduction of the contribution of the industry emissions, but also to a further reduction of the contribution of all other sources.

The large effect of the ozone concentration for option 3 is only the effect of all previous mitigation options. In contrast, if the emissions from industry instead of the land transport emissions are reduced in mitigation option 1, this mitigation would almost have no effect on the ozone concentration. Clearly, the effect of one specific mitigation option strongly depends on the history of previous mitigation options. A combination of tagging and perturbation is a powerful tool for putting additional pressure on unmitigated emission sources, because, even if the absolute ozone levels do not change, their shares in high ozone values (or radiative forcing) increase.

## 5    Analysis of the ozone budget

For more details about the influence of emissions of land transport and ship traffic on the ozone burden, we analysed the burden as well as production and loss rates of $O_3$, $O_3^{tra}$ and $O_3^{shp}$, respectively. These analyses were performed globally, as well as for the distinct geographical regions defined in Sect. 2. Please note, in our tagging method we distinguish only between different emission sources, but not between emission regions. Therefore, the budgets analysed for distinct geographical regions might not be solely influenced by regional emissions, but also by upwind sources.

The global total tropospheric burden of $O_3$ averaged for 2006–2010 is 318 Tg, which is in the range of $337 \pm 23$ Tg presented by Young et al. (2013) as a results of a multi-model intercomparision, but please note that we used a fixed value of 200 hPa for the tropopause. Of these 318 Tg, globally 24 Tg are produced by land transport emissions, while 18 Tg are produced by emissions from shipping. The relative contribution of the burden of $O_3^{tra}$ to the total ozone is thus around 8 % globally and 10 % in the regions Europe, North America and Southeastern Asia. The relative contribution of the burden of $O_3^{shp}$ is around 6 % globally and 8 % near the important source regions. The difference between the rather large contribution of the shipping emissions near ground level (cf. Sect. 3) and the much smaller contribution for the whole troposphere is mainly caused by the confinement of the contribution of shipping emissions to the lowermost troposphere (e.g. Eyring et al., 2007; Hoor et al., 2009).

To better understand the effect of land transport and shipping emissions on the atmospheric composition, we analysed the production and loss rates of $O_3$ from land transport and shipping emissions globally and for the individual regions, respectively. The corresponding numbers are shown in Figs. 7 and 8. Globally integrated production rates of 5274 Tg a$^{-1}$ (averaged 2006–2010) are simulated, while the loss rate is 3972 Tg a$^{-1}$, leading to a net production of $O_3$ of 1301 Tg a$^{-1}$. Similar values of 5110 $\pm$ 606 Tg a$^{-1}$ for the production are reported by Young et al. (2013). The values of the loss are lower than reported by Young et al. (2013), but still within the spread of the different models (4668 $\pm$ 727 Tg a$^{-1}$, again note different definition of the tropopause). Further, it is important to note that loss rates are not calculated consistently in all models presented by Young et al. (2013).

Globally a net production of 165 Tg a$^{-1}$ from the land transport emissions is simulated, corresponding to a contribution of 13 % to the total net $O_3$ production. The contribution of the land transport category to the total net $O_3$ production near the source regions is 19 % over Europe (24 Tg a$^{-1}$), 21 % over North America (39 Tg a$^{-1}$) and 17 % over Southeast Asia (51 Tg a$^{-1}$).

A global net $O_3$ production of emissions from shipping of 129 Tg a$^{-1}$ is simulated, corresponding to a contribution of 10 % to the total net $O_3$ production. Regionally, the importance of the shipping emissions to the net $O_3$ production is much larger. Here contributions of 34 % over the Northern Atlantic (26 Tg a$^{-1}$), 19 % over the Indian Ocean (17 Tg a$^{-1}$) and 52 % over the Northern Pacific (36 Tg a$^{-1}$) are simulated. The larger relative contributions near the source regions compared to the land transport category are mainly caused by less or almost no emissions of other sources in the shipping region. Especially over land, other important sources, such as anthropogenic non traffic and $NO_x$ emissions from soil, decrease the relative importance of the land transport emissions. However, even near the source regions emissions of land transport contribute to around 20 % to the net $O_3$ production in these regions.

## 6  Radiative Forcing

We obtain a global net RF for land transport of $RF_{O3tra}^{tagging} = 92$ mW m$^{-2}$. The shortwave RF is 32 mW m$^{-2}$ and the longwave RF is 61 mW m$^{-2}$. The estimated RF of ship traffic is $RF_{O3shp}^{tagging} = 62$ mW m$^{-2}$ and smaller than the land transport RF. The shortwave RF of ship emissions is 22 mW m$^{-2}$ and the longwave is 40 mW m$^{-2}$. To review estimates of the RF of land transport and shipping emissions and to compare our results with previous estimates, Table 8 compares our results with previous studies. As noted in Sect. 2.3 only the RF of $O_3$ is shown, RF of changes due to $CH_4$ are not considered.

Most studies have estimated a lower RF of land transport/road traffic emissions of around 30 mW m$^{-2}$, using the perturbation approach. The review of Uherek et al. (2010) gives a range for the RF due to road traffic emissions of $50 - (54 \pm 11)$ mW m$^{-2}$. Compared to these values Dahlmann et al. (2011) give larger estimates of around 170 mW m$^{-2}$ using a $NO_x$ only tagging approach and larger global land transport $NO_x$ emissions of roughly 13 Tg(N) a$^{-1}$. Comparing the RF per Tg(N) a$^{-1}$ Dahlmann et al. (2011) reported values of around 14 mW m$^{-2}$ Tg$^{-1}$(N) a, while our estimates are around 10 mW m$^{-2}$ Tg$^{-1}$(N) a.

Also for the RF due to shipping emissions previous estimates using the perturbation approach (around 20–30 mW m$^{-2}$) are lower compared to our findings of around 60 mW m$^{-2}$. Only the tagging study by Dahlmann et al. (2011) report values which are more similar to our estimates (49 mW m$^{-2}$), but this study used lower ship emissions of around 4 Tg(N) a$^{-1}$ while we applied roughly 6 Tg(N) a$^{-1}$. Accordingly, our results suggest a RF of 10 mW m$^{-2}$ Tg$^{-1}$(N) a, while Dahlmann et al. (2011) reported values of around 12 mW m$^{-2}$ Tg$^{-1}$(N) a. Obviously, the NO$_x$ only tagging used by Dahlmann et al. (2011) leads in general to a larger RF per Tg(N) compared to our NO$_x-$ & VOC-tagging.

For a more detailed comparison we also calculated the RF due to land transport and shipping using the 5 % perturbation approach. By this approach we estimate RF$^{\text{perturbation}}_{\Delta \text{O3tra}} = 24$ mW m$^{-2}$ (scaled to 100 %) for land transport emissions and RF$^{\text{perturbation}}_{\Delta \text{O3shp}} = 22$ mW m$^{-2}$ (scaled to 100 %) for shipping emissions. Both values are at the lower end of previous estimates of the RF using the perturbation approach. Remarkable, however, is the difference of a factor of three to four between our results using the perturbation and the tagging approach, despite identical model, emissions, and a consistent calculation of the RF for the impact and the contribution of emissions.

These results have important implications with respect to current estimates of the RF due to land transport (and shipping) emissions. Previous best estimates of an RF of $50 - (54 \pm 11)$ mW m$^{-2}$ by Uherek et al. (2010) are too low, because these estimates are based on the perturbation approach. Previous studies using a NO$_x$-only tagging (Dahlmann et al., 2011; Grewe et al., 2012) reported larger values of up to 170 mW m$^{-2}$, because the NO$_x$-only tagging does not consider competing effects of NO$_x$ and VOCs. Accordingly, our best estimate (92 mW m$^{-2}$) of the RF due to land transport emissions lies between both previous estimates. Compared to this Uherek et al. (2010) gives an estimate of 171 mW m$^{-2}$ of the combined land transport CO$_2$ forcing, while Righi et al. (2015) reports a RF of land transport aerosol in the order of $-81$ to $-12$ mW m$^{-2}$.

The zonal averages of the shortwave, longwave and net radiative forcing for land transport and ship traffic are shown in Fig. 9. Solid (dashed) lines indicate the RF due to the tagging (perturbation) approach. The overall behaviour of RFs deduced by tagging and perturbation approach compare very well. However, the RF obtained by the tagging approach is much larger than the RF obtained by the perturbation approach. In particular, the peak at around 20°N is more enhanced for the tagging approach. This is mainly caused by the larger O$_3$ shares in the upper troposphere, where O$_3$ is most radiative active, as estimated by the tagging compared to the perturbation approach (see Supplement for a figure showing the individual shares). In all cases, the longwave radiative forcing with $\approx 65$ % dominates over the shortwave radiative forcing with $\approx 35$ %. The overall shape of the net forcing corresponds to the tropospheric O$_3^{\text{tra}}$ and O$_3^{\text{shp}}$ column (not shown). In general, the RFs of land transport and ship traffic are largest in the Northern Hemisphere, where most emissions occur. The overall behaviour of the RF zonal means compares quite well with that reported by Myhre et al. (2011), however, we simulate larger absolute values as discussed above.

Figure 10 shows the vertical profile of land transport and ship traffic radiative forcing for the tagging and perturbation approach. Tagging and perturbation approach show the same behaviour. However, the tagging approach has larger values. Most flux changes are simulated in the lower/middle troposphere (300–1000 hPa). Here, the shortwave RF is negative. In contrast, the longwave forcing is positive throughout the whole atmosphere. The vertical profiles correspond to the fraction

of $O_3^{\mathrm{tra}}$ (respectively $O_3^{\mathrm{shp}}$) to $O_3$: the fraction increases with height until it peaks at 850 hPa. In this regime, the largest flux changes occur as well. Above, it continuously decreases with height, so do the flux changes.

## 7 Uncertainties

The general limitations of the tagging diagnostics applied in this study have been discussed by Grewe et al. (2017), therefore we here discuss only the most important details. The mathematical method itself is accurate, but the implementation into the model requires some simplifications such as the introduction of chemical families. Grewe (2004) showed that the implementation of the $NO_y$ family causes an error mainly after the first 12 h after major emission and during this time may lead to an error caused by the family concept of up to 10 %. However, the analyses by Grewe (2004) have only been performed with a simple box model for the upper troposphere and considered only the $NO_y$ family. Applied in an chemistry-climate model this error might be larger, especially with respect to the interplay of freshly emitted lightning-$NO_x$ emissions and oxidized anthropogenic emissions in the upper troposphere. A detailed quantification of this error is difficult. The implementation of the NMHC family causes an additional error, as the different reactivities are not explicitly taken into account. Currently this error cannot be quantified in detail. Other detailed VOC-tagging approaches might help to quantify this error (e.g. Butler et al., 2018). Further, recent updates of the tagging scheme with respect to differences of the $HO_x$ family show an influence of 1–3 percentage-points on the relative contribution of land transport and shipping emissions to ozone (Rieger et al., 2017). In general, we conclude that the error through the simplifications of the tagging method is estimated to be smaller than the errors arising from approximations applied in the global chemistry-climate-models itself (physics and chemistry parameterisations, e.g. 20 % given by Eyring et al., 2007). For the future it would be very interesting to compare results from different tagging methods in more detail to have more quantitative information about the influence of the simplifications chosen by different methods. Other available tagging schemes, however, are based on kinetic approaches (Gromov et al., 2010), consider either only $NO_x$ or VOC (e.g. Emmons et al., 2012; Butler et al., 2011), or are based on thresholds depending on whether the ozone chemistry is $NO_x$ or VOC limited (e.g. Dunker et al., 2002; Kwok et al., 2015). The differences between the assumptions and the scales on which they are applied render a detailed comparison impossible.

However, also the perturbation approach faces an important limitation. The calculated impact largely depends on the magnitude of the chosen perturbation and the impacts are only valid for this specific perturbation (e.g. Hoor et al., 2009). In addition, the perturbation approach has a fundamental problem, namely a non-closed budget. This means that the sum of $O_3$ changes calculated for different perturbed emission sources (e.g. land transport and aviation) is not necessarily the total $O_3$ change if all emissions are reduced at the same time (e.g. Wang et al., 2009; Grewe et al., 2010).

Clearly, the largest sources of uncertainties are the emission inventories. Especially for source attribution not only the uncertainties of the emissions source of interest are important, but also the uncertainties of all other emissions sources. As an example, the emissions of $NO_x$ from soil are poorly constrained (e.g. Vinken et al., 2014). This is in particular problematic as part of the soil-$NO_x$ emissions take place in similar regions as the land transport emissions. Therefore $NO_x$ from both emissions sources influences the ozone production concurrently.

Also with respect to the RF calculation our approach uses some assumptions (for the tagging and the perturbation results, respectively) which we discuss in detail in Sect. 2.3 and the Supplement. Further, due to the large sensitivity of the RF to ozone in the upper troposphere in particular lightning-$NO_x$ shows a large radiative efficiency (Dahlmann et al., 2011) errors in the attribution due to the $NO_y$ family approach (see above) can lead here to an overestimated RF. This needs to be investigated in more detail in the future. Compared to calculations of the ozone radiative forcing by comparing two simulation results applying conditions for present day and preindustrial times we estimate a difference of of 10–30 % (for details see Supplement). In general, these differences are smaller as the factor 2–3 between the results of the tagging and the perturbation approach.

## 8   Summary and Conclusion

We estimate the contribution of land transport and shipping emissions to tropospheric ozone for the first time with an advanced tagging method which considers not only $NO_x$, but also CO and VOC. Our results indicate a maximum contribution of land transport emissions during summer of up to 18 % to ground level ozone in North America and 16 % in Southern Europe, which corresponds to up to 12 $nmol\,mol^{-1}$ in North America and 10 $nmol\,mol^{-1}$ in Europe.

The largest contribution of shipping emissions to ground level ozone was simulated in the Northern Pacific Ocean and the Northern Atlantic Ocean. During summer, contributions of up to 30 % were simulated in the Northwestern Pacific Ocean, corresponding to up to 12 $nmol\,mol^{-1}$. In the Northern Atlantic Ocean contributions of up to 20 % during summer were calculated (up to 12 $nmol\,mol^{-1}$). The comparison with previous estimates clearly show that the results strongly depend on the chosen method. Perturbation studies using a 5 % approach usually show the lowest contribution (scaled to 100 %) in the considered regions, while most 100 % perturbations, as well as the tagging approach show the largest contributions.

Overall, emissions of land transport and ship traffic contribute by 8 % and 6 %, respectively, to the tropospheric ozone burden. Land transport emissions contribute by around 20 % to the tropospheric ozone production near the source regions. The contribution of shipping emissions to the net ozone production near the source regions is with values of up to 52 % in the Northern Pacific even larger as the contribution of land transport emissions to the net production.

Using the tagging method we estimate a global average radiative forcing due to ozone caused by land transport emissions of 92 $mW\,m^{-2}$ and 62 $mW\,m^{-2}$ caused by shipping emissions. In general, radiative forcings are largest on the Northern Hemisphere and peak at around $30°$ N. While our estimates of the contribution of land transport and shipping emissions to tropospheric ozone are similar compared to previous studies using a 100 % perturbation, our estimates of the radiative forcing are larger by a factor of 2–3 compared to previous estimates using the perturbation method. As discussed in detail, this large difference compared to previous values is largely attributable to differences in the methodology, leading to different estimates of the ozone shares attributable to land transport and shipping emissions, respectively. Previous estimates of the ozone RF due to land transport emissions using a $NO_x$-only tagging method, however, are too large as they do not consider the competing effects of $NO_x$ and VOCs. Accordingly, 92  and 62 $mW\,m^{-2}$ are the current best estimates of the ozone RF due to land transport and shipping emissions, as estimated using a source apportionment method.

Our results clearly indicate that it is important to differentiate between sensitivity methods (i.e. perturbation), which estimate the impact, and the source apportionment methods (i.e. tagging) which estimate the contribution of emissions, because both approaches give answers to different questions. The perturbation approach measures the effect of an emission change, while only the tagging approach yields contributions of individual emission sources to ozone concentration. This difference is very important when interpreting the results, in particular when investigating the radiative forcing of individual emission categories. To investigate mitigation options, the tagging method cannot replace sensitivity (i.e. perturbation) studies and vice versa. However, we demonstrated that even if mitigation options are investigated, the sensitivity simulations should be equipped with a tagging method. The tagging approach provides very valuable additional information about the changes of the contributions to ozone due to the mitigation option, which puts additional pressure on unmitigated sources.

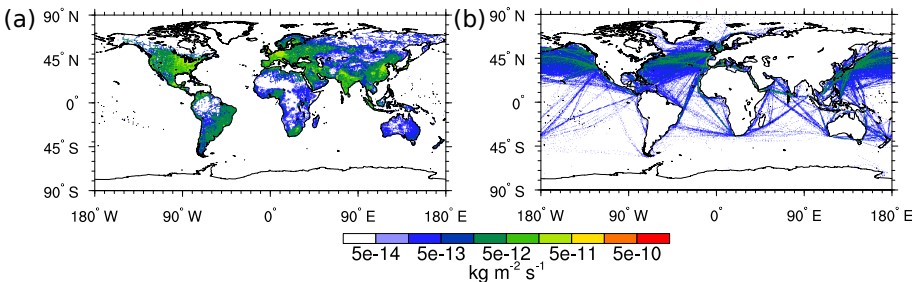

**Figure 1.** Average (2006–2010) emissions flux of $NO_x$ (in $kg(N)\ m^{-2}\ s^{-1}$) emissions from **(a)** land transport and **(b)** shipping.

**Table 1.** Description of the different categories as used by the TAGGING submodel.

| tagging categories | description |
|---|---|
| land transport | emissions of road traffic, inland navigation, railways (IPCC code 1A3b_c_e) |
| anthropogenic non-traffic | sectors Energy, Solvents, Waste, Industries, Residential, Agriculture |
| ship | emissions from ships (IPCC code 1A3d) |
| aviation | emissions from aircraft |
| lightning | lightning $NO_x$ emissions |
| biogenic | on-line calculated isoprene and soil-$NO_x$ emissions, off-line emissions from biogenic sources and agricultural waste burning (IPCC code 4F) |
| biomass burning | biomass burning emissions |
| $CH_4$ | degradation of $CH_4$ |
| $N_2O$ | degradation of $N_2O$ |
| stratosphere | downward transport from the stratosphere |

**Table 2.** Average (2006–2010) annual total emission of CO (in $Tg(CO)\,a^{-1}$), $NO_x$ (in $Tg(N)\,a^{-1}$) and NMHC (in amount of carbon) of the most important emission categories. The category 'other' contains the emissions of the sectors biomass burning, agricultural waste burning as well as other biogenic emissions.

| | CO $(Tg(CO)\,a^{-1})$ | NMHC $(Tg(C)\,a^{-1})$ | $NO_x$ $(Tg(N)\,a^{-1})$ |
|---|---|---|---|
| land transport | 152 | 17 | 10 |
| shipping | 1 | 2 | 6 |
| anthropogenic non-traffic | 411 | 73 | 17 |
| soil NOx | | | 6 |
| lightning NOx | | | 5 |
| biogenic $C_5H_8$ | | 493 | |
| other | 416 | 15 | 5 |

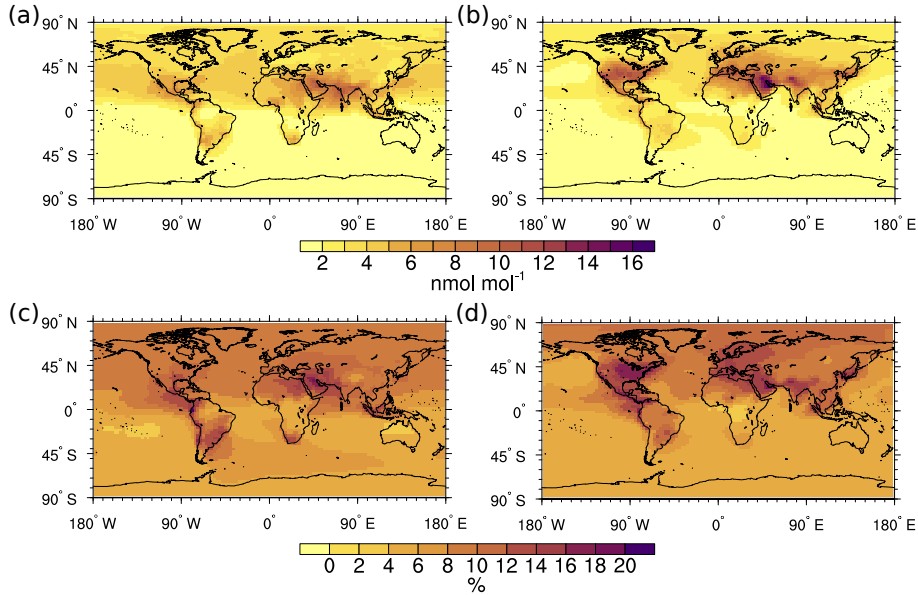

**Figure 2.** Seasonal average values of the absolute and relative contribution of $O_3^{tra}$ to near ground level $O_3$. The upper row give the absolute values (in $nmol\,mol^{-1}$) for winter (DJF, **(a)**) and summer (JJA, **(b)**), respectively. The lower row shows the DJF **(c)** and JJA **(d)** values of the contribution (in %).

**Table 3.** Summary of previous global model studies investigating the contribution/impact of land transport/road traffic emissions to ozone. Method denotes the percentage of the emissions reductions (perturbation). The other columns list the amount of land transport/road traffic emissions as well as the fraction (f) compared to the emissions used in the studies for $NO_x$ (in Tg(N) $a^{-1}$), CO (in Tg(CO) $a^{-1}$) and NMHC (Tg(C) $a^{-1}$). The four rows from the right list the contribution of the land transport/road traffic categories as estimated by these studies in mixing ratios and/or percent. Where possible, we show the estimated contribution for the geographical regions defined in Sect. 2 as well as zonal average values. All contributions are given to near ground level ozone and for July conditions. The table is ordered by the year of publication. A '-' indicates missing information.

| study | method | $NO_x$ | $fNO_x$ | CO | fCO | NMHC | fNMHC | NA | EU | SEA | ZM |
|---|---|---|---|---|---|---|---|---|---|---|---|
| | | | | | | | | nmol mol$^{-1}$ | nmol mol$^{-1}$ | nmol mol$^{-1}$ | nmol mol$^{-1}$ |
| | | Tg a$^{-1}$ | % | Tg a$^{-1}$ | % | Tg a$^{-1}$ | % | % | % | % | % |
| GB03 | 100% | 10 | 24 | 207 | 14 | - | - | - | - | - | - |
| | | | | | | | | 11–15 | 9–15 | 5–12 | - |
| NM06 | 100% | 9 | 30[a] | 196 | 36[a] | 36 | 27[a] | 5–20 | 5–15 | 5–10 | - |
| | | | | | | | | 10–50 | -5–25 | 5–50 | - |
| NM06 | 100% | 9 | 30[a] | 196 | 36[a] | 36 | 27[a] | zonal mean | | | - |
| | | | | | | | | | | | up to 10 |
| M07 | 100% | 9 | 24 | 237 | - | 27 | 5 | - | - | - | - |
| | | | | | | | | 13–16 | 9–16 | 3–16 | - |
| M07 | 100% | 9 | 24 | 237 | - | 27 | 5 | zonal mean | | | up to 5 |
| | | | | | | | | | | | up to 12 |
| H09 | 5 %[b] | 7 | 15 | 31 | 7 | 8 | 2 | 2–5[c] | 2–6[c] | 1–4[c] | - |
| | | | | | | | | - | - | - | - |
| K10 | 5 %[b] | 9 | 18 | 110 | 11 | 11 | 1 | 2–5 | -1–5 | 1–3 | - |
| | | | | | | | | - | - | - | - |
| K10 | 100 % | 9 | 18 | 110 | 11 | 11 | 1 | zonal mean ground level | | | - |
| | | | | | | | | | | | up to 7 |
| this study | tagging | 10 | 20 | 152 | 16 | 17 | 3 | 3–14 | 3–13 | 2–11 | |
| | | | | | | | | 6–19 | 8–18 | 5–16 | |
| this study | tagging | 10 | 20 | 152 | 16 | 17 | 3 | zonal mean mid latitudes NH | | | 3–7 |
| | | | | | | | | | | | 9–11 |
| this study | 5 %[b] | 10 | 20 | 152 | 16 | 17 | 3 | 1–9 | -1–6 | -1–5 | - |
| | | | | | | | | 1–12 | -3–9 | -2–12 | - |
| this study | 5 %[b] | 10 | 20 | 152 | 16 | 17 | 3 | zonal mean mid latitudes NH | | | 2–4 |
| | | | | | | | | | | | 1–2 |

[a] Fraction only compared to all anthropogenic emissions. [b] Given values scaled to 100 %. [c] Given for average values from 800 hPa to the surface.

Abbreviations are: GB03 (Granier and Brasseur, 2003), N06 (Niemeier et al., 2006), M07 (Matthes et al., 2007), H09 (Hoor et al., 2009), K10 (Koffi et al., 2010).

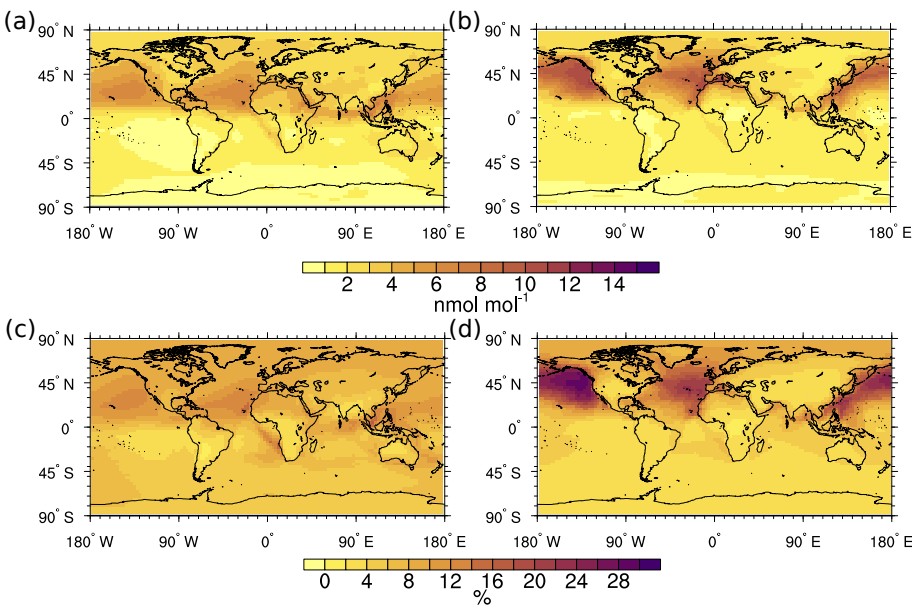

**Figure 3.** Seasonal average values of the absolute and relative contribution of $O_3^{shp}$ to near ground level $O_3$. The upper row give the absolute values (in $\mathrm{nmol\ mol^{-1}}$ for DJF **(a)** and JJA **(b)**, respectively. The lower row shows the DJF **(c)** and JJA **(d)** values of the contribution (in %).

**Table 4.** Summary of previous global model studies investigating the contribution/impact of shipping emissions to ozone. Method denotes the percentage of the emissions reductions (perturbation). The other columns list the amount of shipping emissions as well as the fraction (f) compared to all emissions used in the studies for $NO_x$ (in $Tg(N)\ a^{-1}$ ). The four rows from the right list the contribution of the shipping category as estimated by these studies in mixing ratios (upper row) and/or percent (lower row). Where possible, we show the estimated contribution for the geographical regions defined in Sect. 2 as well as zonal average values. For the geographical regions we give only the values larger than the background values. All contributions are given to near ground level ozone and for July conditions. The table is ordered by the year of publication. A '-' indicates missing information.

| study | method | $NO_x$ | $fNO_x$ | Atlantic $nmol\ mol^{-1}$ | Pacific $nmol\ mol^{-1}$ | India $nmol\ mol^{-1}$ | Zonal Mean $nmol\ mol^{-1}$ |
|---|---|---|---|---|---|---|---|
| | | $Tg\ a^{-1}$ | % | % | % | % | % |
| ED03 | 100% | 4 | 8 | 4–12 | 4–11 | 3–4 | - |
| | | | | - | - | - | - |
| E07 | 100% | 3 | 11[a] | 2–12 | 1–4 | 1–4 | - |
| | | | | 12–36 | 12–24 | 12–18 | - |
| E07 | 100% | 3 | 11[a] | zonal mean mid latitude NH | | | 1–1.5 |
| | | | | | | | - |
| H09 | 5%[c] | 4 | 10 | 2–4 | 2–3 | 1–2 | - |
| | | | | - | - | - | - |
| D09 | 100 % | 5 | - | - | - | - | - |
| | | | | 14–33 | 14–40 | 9–12 | - |
| K10 | 5%[c] | 4 | 8 | 2–5 | 3–6 | 1–2 | - |
| | | | | - | - | - | - |
| K10 | 5%[c] | 4 | 8 | zonal mean | | | up to 1.5 |
| | | | | | | | - |
| K10 | 100% | 4 | 8 | up to 8 | up to 9 | - | - |
| | | | | - | - | - | - |
| K10 | 100% | 4 | 8 | zonal mean | | | up to 3 |
| | | | | | | | - |
| this study | tagging | 6 | 12 | 3–9 | 4–11 | 2–5 | - |
| | | | | 10–24 | 10–33 | 9–15 | - |
| this study | tagging | 6 | 12 | zonal mean mid latitudes NH | | | 3–6 |
| | | | | | | | 10–15 |
| this study | 5 %[c] | 6 | 12 | 2–8 | 2–7 | 1–4 | - |
| | | | | 10–18 | 11–22 | 4–10 | - |
| this study | 5 %[c] | 6 | 12 | zonal mean mid latitudes NH | | | 2–4 |
| | | | | | | | 5–8 |

[a] No information available. [b] Fraction only compared to all anthropogenic emissions. [c] Given values scaled to 100 %. [d] Given for average values from 800 hPa to the surface. Abbreviations are: ED03 (Endresen et al., 2003), E07 (Eyring et al., 2007), H09 (Hoor et al., 2009),D09 (Dalsøren et al., 2009), K10 (Koffi et al., 2010).

**Table 5.** Comparison of Γ values (definition see text) between the four considered regions and interpretation of these values.

|  | Γ | Interpretation |
|---|---|---|
| Europe | 0.9 | 90 % of the $O_3$ reduction due to land transport emissions are compensated by increased ozone production. Ozone contribution and impact differ largely. |
| Southeast Asia | 0.6 | 10 % reduction of land transport emissions will lead to a 4 % reduction in ozone due to increased ozone productivity. Ozone contribution and impact differs largely. |
| North Africa | 0.4 | Only small compensation effects; ozone contribution and impact differ only slightly. |
| South America | 0.3 | Land transport emission reduction almost scales with ozone reduction. Impact and contribution are almost equal. |

**Table 6.** Burden of $O_3$ and $O_3^{tra}$ integrated up to 200 hPa (in Tg). Average values for the period 2006–2010.

|  | $O_3$ (Tg) | $O_3^{tra}$ (Tg) | contribution $O_3^{tra}$ (%) |
|---|---|---|---|
| Global | 318 | 24 | 8 |
| Europe | 15 | 2 | 10 |
| North America | 21 | 2 | 10 |
| Southeast Asia | 25 | 2 | 9 |

**Table 7.** Burden of $O_3$ (total) and $O_3^{shp}$ (shipping) integrated up to 200 hPa (in Tg). Average values for the period 2006–2010.

|  | $O_3$ (Tg) | $O_3^{shp}$ (Tg) | contribution $O_3^{shp}$ (%) |
|---|---|---|---|
| Global | 318 | 18 | 6 |
| North Atlantic Ocean | 24 | 2 | 8 |
| Indian Ocean | 27 | 1 | 5 |
| North Pacific Ocean | 32 | 2 | 8 |

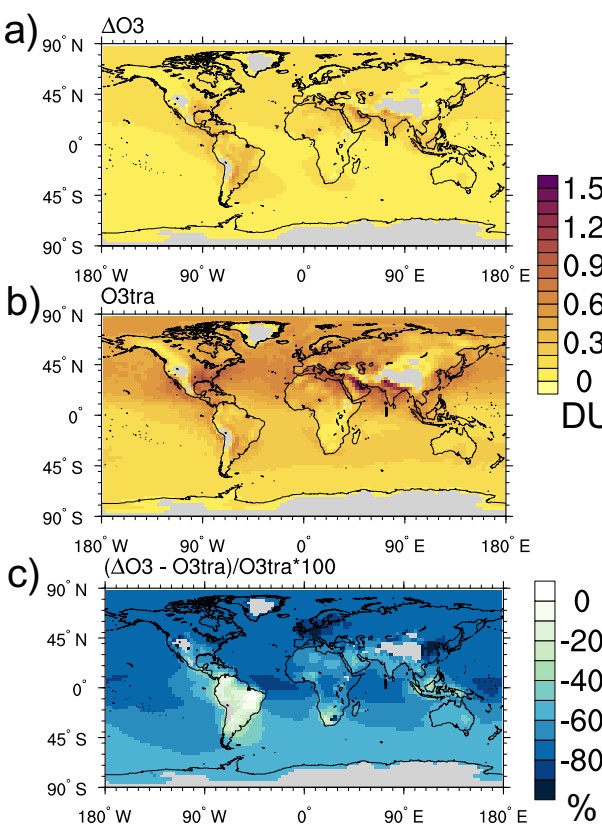

**Figure 4.** Multi-annual averages (2006–2010) of **(a)** $\Delta O_3$ (impact), **(b)** $O_3^{tra}$ (contribution, both in DU) of the *REF* simulation and **(c)** the relative difference between the impact and the contribution of land transport emissions (in %). All values are calculated for the partial columns from the surface of up to 850 hPa (850PC).

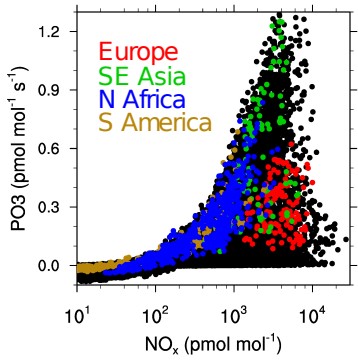

**Figure 5.** Dependency between $NO_x$ mixing ratios and net $O_3$ production. The black dots represent monthly mean values at ground level for the year 2010 of every individual grid box. The individual colours indicate monthly average values during May–August (Northern Hemisphere) and November–February (Southern Hemisphere) for individual regions (defined as rectangular areas).

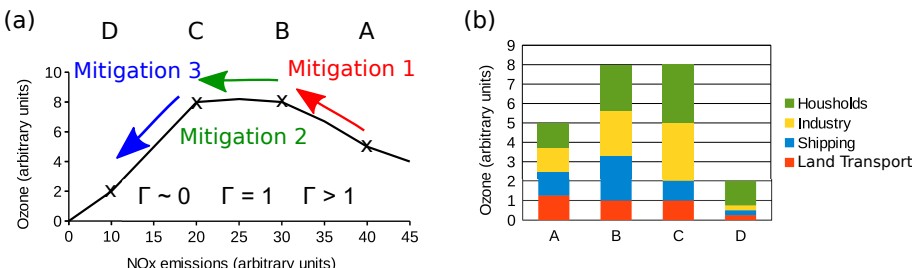

**Figure 6.** Idealised example explaining the difference of the perturbation and the tagging approach for the evaluation of mitigation increases. (a) shows the dependency between $NO_x$ emissions and ozone (both in arbitrary units). Three different mitigation options are indicated by the colored arrows. In addition, the approximate value of $\Gamma$ (see text for definition) is given. (b) shows the contribution of the ozone concentration at the four marked points in (a). In this example it is assumed that only four emission categories exist, emitting the same amount of emissions at point A.

**Table 8.** Global estimates of the annually averaged radiative forcing due to $O_3$ caused by emissions of land transport/road traffic (global RF road) and ship emissions (global RF shp). Please note that individual studies use different methods for the calculation of the radiative forcing e.g. some studies give instantaneous values, while other studies stratospheric adjusted values (see last row).

| Study | method | global RF road (mW m$^{-2}$) | global RF shp (mW m$^{-2}$) | RF type |
|---|---|---|---|---|
| Endresen et al. (2003) | 100 % | - | 29 | scaling of tropospheric ozone column change |
| Niemeier et al. (2006) | 100 % | 30 / 50 (January / July) | - | instantaneous at TP[e] |
| Eyring et al. (2007) | 100 % | - | $10 \pm 2$ | instantaneous at TP[e] decreased by 22 % |
| Fuglestvedt et al. (2008) | 100 % | $54 \pm 11$ | $32 \pm 9$ | stratospheric adjusted |
| Hoor et al. (2009) | 5 % | 28[a] | 28[a] | - |
| Uherek et al. (2010) | review | $50 - (54 \pm 11)$ | - | - |
| Dahlmann et al. (2011) | $NO_x$-tagging | 170[c] | 49[c] | fixed dynamical heating |
| Dahlmann et al. (2011) | 100 % | 31[c] | - | fixed dynamical heating |
| Myhre et al. (2011) | 5 % | 31[a] | 24[a] | - |
| Grewe et al. (2012) | $NO_x$-tagging | 132[c] | - | fixed dynamical heating |
| Grewe et al. (2012) | 100 % | 24[c] | - | fixed dynamical heating |
| Holmes et al. (2014) | 5 % | - | 27[d] | - |
| this study | $NO_x$/VOC-tagging | 92 | 62 | stratospheric adjusted |
| this study | 5 % | 24[a] | 22[a] | stratospheric adjusted |

[a] Scaled to 100 %. [b] For year 2000 conditions. [c] For year 1990 conditions. [d]Calculated by scaling the RF value of the 'instant dilution' case for a change of 1 Tg a$^{-1}$ with the total amount of used emissions by Holmes et al. (2014). [e] Tropopause

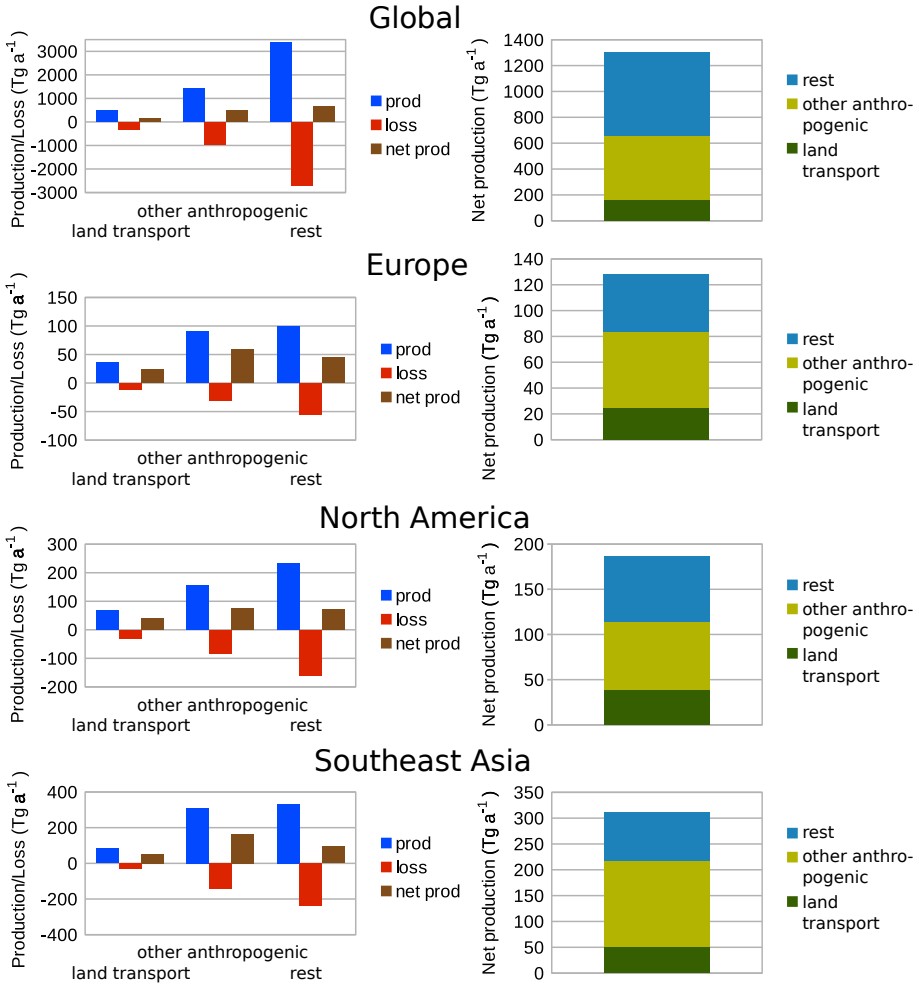

**Figure 7.** Production and loss rates of $O_3$ from different sectors (integrated up to 200 hPa and averaged for 2006–2010). The left side shows the individual production and loss rates as well as the net $O_3$ production, while the right side shows only the net production of the different sectors. For simplicity only land transport, other anthropogenic (shipping, anthropogenic non-traffic and aviation) and rest (all other tagging categories) are shown.

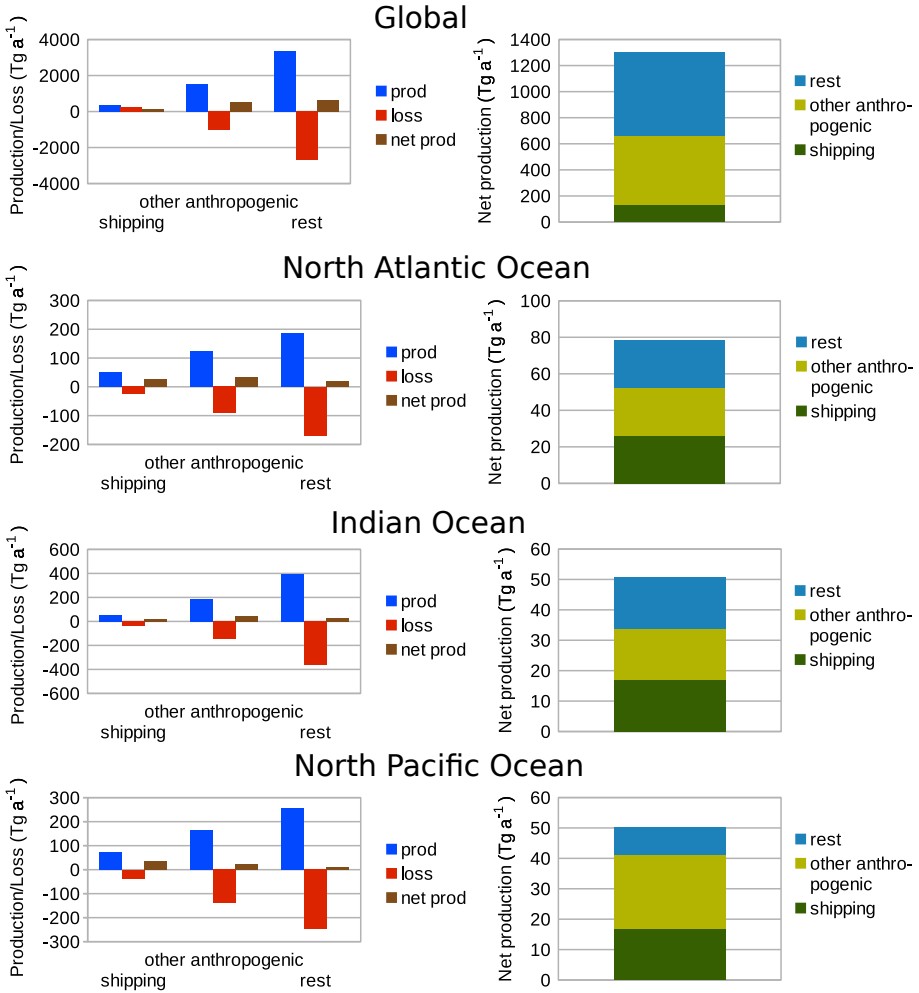

**Figure 8.** Production and loss rates of $O_3$ from different sectors (integrated up to 200 hPa and averaged for 2006–2010). The left side shows the individual production and loss rates as well as the net $O_3$ production, while the right side shows only the net production of the different sectors. For simplicity only shipping, other anthropogenic (land transport, anthropogenic non-traffic and aviation) and rest (all other tagging categories) are shown.

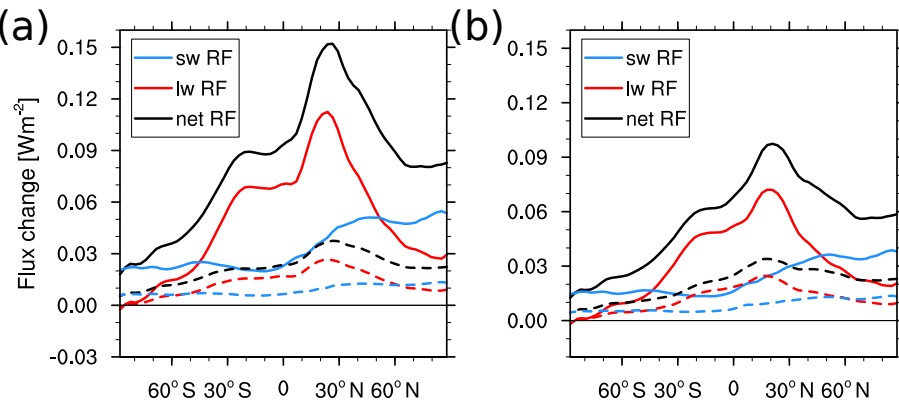

**Figure 9.** Zonal mean of shortwave, longwave and net radiative $O_3$ forcing of (a) land transport and (b) ship traffic. The continuous lines give the results of the tagging method, the dashed lines of the perturbation method.

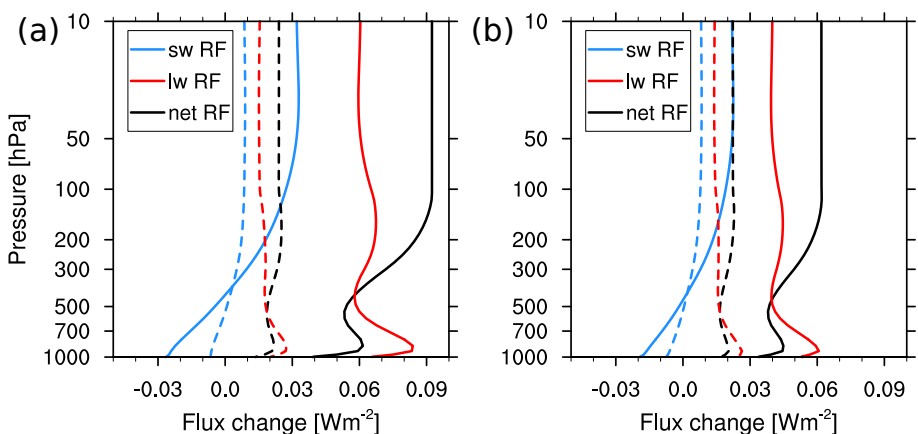

**Figure 10.** Vertical profile of globally averaged shortwave, longwave and net radiative $O_3$ forcing of (a) land transport and (b) ship traffic. The continuous lines give the results of the tagging method, the dashed lines of the perturbation method.

*Acknowledgements.* M.Mertens acknowledges funding by the DLR project 'Verkehr in Europa' and 'Auswirkungen von $NO_x$'. Furthermore, part of this work is funded by the DLR internal project 'VEU2'. We thank R. Sausen, M. Righi (both DLR) and two anonymous reviewer which improved this manuscript. Analysis and graphics of the used data was performed using the NCAR Command Language (Version 6.4.0) Software developed by UCAR/NCAR/CISL/TDD and available on-line: http://dx.doi.org/10.5065/D6WD3XH5. Computational resources

5   for the simulation were provided by the German Climate Computing Centre (DKRZ) in Hamburg (project 0617).

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
