# Peer review of "Revisiting the contribution of land transport and shipping emissions to tropospheric ozone"

_Atmospheric Chemistry and Physics, 2017_

## Referee Comment (RC1) · Anonymous Referee #1 · 9 Nov 2017

Title: Revisiting the contribution of land transport and shipping emissions to tropospheric ozone.

Overview: This paper estimates contributions to ozone using a tagging methodology. They focus on land transportation and shipping, which are important sectors. They compare their results to comparable studies from the past and attempt to distinguish between perturbation and "contributions." The methods are generally clear and the results are well presented. There are several points of interpretation and extension of this work to conclusions that go beyond what the work supports. The main problem in this paper is cooption of terms that this reviewer believes are inappropriate. Much of

this is framing, but has important implications that need to be better fleshed out.

The field has historically estimated "contribution" in many ways including perturbation, source apportionment tagging (e.g., CAMx OSAT/APCA and CMAQ ISAM), renormalized sensitivities (e.g., DDM or adjoint). Yet this paper argues that "only tagging estimates the contribution of emissions." Note that many tagging techniques (OSAT/APCA and ISAM) have sensitivity-based metrics to account for relative importance (e.g., Sillman-ratio threshold). One goal of the relative importance approaches is to make a "contribution" that is meaningfully consistent with sensitivity because of its usefulness to policy makers. These relative importance factors are omitted in the technique applied in this paper. Why is this combinatorial tagging the only approach that can estimate "contribution"? If combinatorial tagging is somehow more appropriate, then why not include all reactants? The ad absurdum argument would then say that a large fraction of all ozone is simply natural due to molecular oxygen required for the formation of RO2. Thus, the formulation already assumes that limiting factors are important. Why is the limiting factor not important between NOx and VOC in "contribution"?

The IPCC AR5 WG1 Chapter 8 defined radiative forcing as "an instantaneous change in net (down minus up) radiative flux (shortwave plus longwave; in W m–2) due to an imposed change." AR5s definition is generally consistent with previous definitions (e.g., Seinfeld and Pandis 2006; Jacob 1999). Contribution as defined as the combinatorial tagging is not consistent with an imposed change. First, there is no imposed change. In fact, removing those emissions (tra or shp) would not impose a change of similar magnitude. Thus, the idea that transport or shipping contributes to RF proportionally to combinatorial tagging is conceptually flawed.

The authors assert that this technique is useful in understanding changes in emissions (particularly section 4.1). The current state of practice uses an emission reduction matrix to explore sensitivities at multiple emissions reductions (20, 40, 60, 80%) of both NOx and VOC. How is tagging this technique more useful than the iterative NOx/VOC matrix?

Finally, I have concern about the methodology as described in Eq 2. Apportionment based on fraction of NOy and NMHC concerns me. See Page,Line comments.

Much of this critique is specific to the interpretation and assertions of unique value. The methods and results are internally consistent. I am skeptical of the species family approach as described. The biggest issue is that the article attempts to fully own the term "contribution", applies combinatorial tagging to RF in an odd way that needs to be clearly distinguished from traditional RF, and implies regulatory value that is likely already met. Most of these comments can be addressed by revising the interpretation.

Page,Line Comments:

1,3: recommend "complementary" because the dynamics of "competition"

1,5-7: The regions are not clear in the abstract. Consider adding "ocean" to each region to be consistent with text and clarify.

1,20: This is a narrow definition of the word contribution and I have seen no argument that combinatorial tagging is the only way to define contribution.

2,15: It is not important to know "contribution" as defined by combinatorial tagging to define mitigation strategies. In fact, knowing sensitivity is fundamentally more important to mitigation since the mitigation intends to impose a change.

3,4: F should be f

4,23-5,2: If implemented as discussed, this approach assumes two things that are fundamentally at odds with our understanding of atmospheric chemistry. - First, it assumes that all NOy (NOx + NOz) is equally available for ozone production. This is problematic because NOy photochemical lifetime is much longer than NOx. As a result, this Eq 2 will attribute ozone production to NOx and NOz proportionally. That would lead to ozone being attributed to HNO3^tag in the mid to upper troposphere. Unless NOy is being defined differently than the field convention, this is troubling. - Second, and less concerning, NMHC are not all equally reactive nor do they have equal RO2

yields. Assuming concentration fractions are proportional to combinatorial contribution is not consistent with the chemical mechanism.

5,23: Februar[y]

6,12: Is the seasonality of non-traffic reasonable and expected?

6,24: Why is July most comparable? What did those studies look at?

7,3: Reword or edit grammar

7,8: This assumes that contribution == tagging, which the authors need to further consider.

9,1-4: Are these ratios of partial column or average ratios?

9,7: contribution...

10,2: consider replacing "almost" with "closest to".

10,7-12: Are not mitigation strategies more aligned with sensitivities?

10,20-31: See discussion of sensitivity matrix, which is the current approach for developing mitigation.

11,20: "[global] land transport." This section is tricky because the production may come from upwind sources. Try to be more explicit.

13,24: Be more specific than "some".

13,25: trough -> through?

13,25: is the author referring to engineering simplifications in the CTM?

13,28-29: CAMx OSAT/APCA[camx.com] and CMAQ ISAM [doi: 10.5194/gmd-8-99-2015] are a couple of examples of similar complexity to this scheme.

14,22-24: One interpretation is that the radiative forcing in this paper is an overestimate

due to the lack of realism in the tagging compared to an actual imposed response.

---

## Referee Comment (RC2) · Anonymous Referee #2 · 19 Dec 2017

This paper offers a nice overview of the impact of shipping emissions on ozone through the use of two methodologies: the tagging methodology and the perturbation methodology. The paper is well written and extremely thorough with a clear comparison to previous studies.

Comments.

1. The authors state two goals in this study (p3, l4-6) in determining ozone from shipping emissions: to review previous studies and to give the results of the tagging method. The results from the authors use of the tagging methodology nicely complements estimates from the contribution method. I think the paper works as a review

paper. However, as written, I question whether the paper stands very well on its own as a new piece of research. There does not seem to be enough new. Part of the author's justification for this paper is that no one has investigated the ozone contributions from transportation using the tagging approach. Just because something has not been done does not mean it is scientifically interesting or worth pursuing. There are probably other emission sectors that have not been investigated using the tagging approach. It doesn't seem that there should be a new paper written for each of these sectors. I think the authors need to better justify their study than simply state it has not been done. Why do we need another paper on the emission contributions from transportation emissions given the uncertainty? Specifically, what new insights does the tagging approach give? (This needs to be better clarified, see below). What do we learn about the tagging approach here that we didn't know before?

2. Why does the present study use a 5% perturbation? The results are sensitive to this perturbation. Some justification is needed. It would be helpful for comparison purposes if the authors also gave their results for a 100% perturbation in their tables. To what extent does the discrepancy with the tagging method come from the assumed 5% reduction? It appears a 100% emission reduction gives similar results to the tagging method. Reporting on a 100% emission perturbation would also help compare with other studies.

3. Equation (3): Is a factor of 20 missing?

4. The definition of gamma needs to be clarified in more detail in the text. After looking in detail at the figure and reading the text the meaning of gamma became clear, but it should not have been this difficult. Please clarify the definition of gamma in the text explicitly stating what the y intercept is and stating that y is the average net ozone production rate in a particular region.

5. It is unclear why gamma is defined in terms of the intercept instead of the slope (dO3/dNOx). The intercept will be leveraged by the amount of the NOx emissions.

That is, the impact of the slope the will be amplified when the NOx emissions are large by changing the intercept to a greater extent than if the NOx emissions are small.

6. Figure 5 clearly demonstrates that the perturbation approach gives different estimates under different conditions. However, it does not show how the tagging approach differs. Some more work is needed here to better understand how these two approaches give different answers depending on ambient conditions and transportation emissions. From line 9 onwards (on page 9) the well-known dependence of ozone production on NOx is shown, with the well-known result that in regions of high NOx a decrease in emissions has little impact on the ozone concentration. There is not much new here. The text and figures don't explicitly show that the tagging approach gives a different answer than the perturbation approach. And isn't the discrepancy between the two methods well known. What is new?

7. The authors state: "Combining the tagging and the perturbation approach is therefore the best way to measure the success of a mitigation strategy." (p10, l19-20). The authors argue that the perturbation approach gives different answers depending on the current state. I suppose the tagging approach gives the same ozone reduction regardless of the mitigation pathway. This should be clearly stated. Nevertheless, it is unclear how one would use the tagging method to decide on mitigation issues. Perhaps a concrete example would be helpful here? This is important because it would provide a needed justification of the tagging approach. It is crucial that the paper clearly gets this across. It seems to me the tagging approach is useful in assigning blame: for example, if you want to apportion blame for an ozone pollution outbreak or for the radiative forcing due to ozone. It is not clear to me how one would use the tagging method practically in assessing mitigation options.

8. The loss rate of ozone is very dependent on how it is calculated (page 11). How are the losses calculated in the present study? Are they calculated in the same manner in the comparison studies?

9. P12, l 16: "We obtain...." Using which method?

10. The section on uncertainties should also discuss the uncertainties in the perturbation method. In particular this method is sensitive to the perturbation assumed.
* * *

---

## Author Comment (AC1) · 16 Feb 2018

We thank referee#1 for many useful comments which helped to improve the manuscript. In the following, referee comments are given in italics, our reply's in normal font, and text passages which we included in the text, in bold.

*Overview: This paper estimates contributions to ozone using a tagging methodology. They focus on land transportation and shipping, which are important sectors. They compare their results to comparable studies from the past and attempt to distinguish between perturbation and "contributions." The methods are generally clear and the results are well presented. There are several points of interpretation and extension of this work to conclusions that go beyond what the work supports. The main problem in this paper is cooption of terms that this reviewer believes are inappropriate. Much of this is framing, but has important implications that need to be better fleshed out.*

Reply: We thank referee#1 for these positive comments. We modified the text accordingly and described the terms we use in more detail or changed parts which might be misleading. Please see below for more detailed responses.

*The field has historically estimated "contribution" in many ways including perturbation, source apportionment tagging (e.g., CAMx OSAT/APCA and CMAQ ISAM), renormalized sensitivities (e.g., DDM or adjoint). Yet this paper argues that "only tagging estimates the contribution of emissions." Note that many tagging techniques (OSAT/APCAvand ISAM) have sensitivity-based metrics to account for relative importance (e.g., Sillman-ratio threshold). One goal of the relative importance approaches is to make a "contribution" that is meaningfully consistent with sensitivity because of its useful- ness to policy makers. These relative importance factors are omitted in the technique applied in this paper. Why is this combinatorial tagging the only approach that can estimate "contribution"? If combinatorial tagging is somehow more appropriate, then why not include all reactants? The ad absurdum argument would then say that a large fraction of all ozone is simply natural due to molecular oxygen required for the formation of RO2. Thus, the formulation already assumes that limiting factors are important. Why is the limiting factor not important between NOx and VOC in "contribution"?*

Reply: We agree with referee#1 that in the past the term 'contribution' has been used for the results of different methods. However, in the last years this difference between 'impact' (sensitivity, e.g. perturbation or DDM) and 'contribution' (source apportionment, e.g. tagging) has been discussed in several publications from both, the chemistry-climate, and air quality communities (e.g. Grewe et al., 2010; Clappier et al., 2017). Of course, large differences between various source apportionment methods exits, some consider $NO_x$ or VOC only (e.g. Grewe, 2004; Emmons et al., 2012), our method considers $NO_x$ and VOCs, others use thresholds to judge, whether the chemistry is $NO_x$ or VOC limited and attribute ozone to $NO_x$ or VOC emission sources (e.g. Dunker et al., 2002; Kwok et al., 2015).

The calculated contribution of course heavily depends on the applied source apportionment methods. We don't want to judge on any of these approaches

being right or wrong. However, the contributions calculated using a 'NO$_x$ or VOC limit'-threshold are by definition more sensitivity based and not comparable to the contribution estimated by considering NO$_x$ and VOC only, or together.

Our goal was not to say that only the combinatorial tagging can be used to calculate contributions. But the general difference between these source apportionment methods, which usually have closed budgets, and the sensitivity methods is important to us. We revised large parts of the Introduction (see also reply to referee#2) to make clear that we separate between impact/contribution and sensitivity/source apportionment.

The most important change with respect to this comment is:

**With respect to the influence of different emission sources on ozone itself, typically two different questions are of interest (e.g. Wang et al., 2009; Grewe et al., 2010; Clappier et al., 2017):**

- **How sensitive does ozone respond to changes of a specific emission source (sensitivity study)?**

- **How large is the contribution of different emission sources to ozone (source apportionment)?**

**Sensitivity studies are important to investigate the influence of an emission change on, for instance, ozone. Often, the so called perturbation approach has been applied, in which the results of two (or more) simulations are compared: one reference simulation with all emissions and a sensitivity simulation with perturbed emissions. Source apportionment, in contrast, is important to attribute different emission sources to climate impact (such as radiative forcing) or extreme ozone events. Source apportionment studies often use tagged tracers in order to estimate contributions of different emission sources, for instance, to ozone. In this tagging approach, additional diagnostic species are introduced, which follow the reaction pathways of the emissions from different sources (e.g. Lelieveld and Dentener, 2000; Dunker et al., 2002; Grewe, 2004; Gromov et al., 2010; Butler et al., 2011; Grewe et al., 2012; Emmons et al., 2012; Kwok et al., 2015). Other methods exist for both type of studies, which we neglect here for simplicity (see e.g. Clappier et al., 2017).**

**In a linear system, both approaches, perturbation and tagging, lead to the same answer (e.g. Grewe et al., 2010; Clappier et al., 2017). The O$_3$ chemistry, however, is highly non-linear. Therefore, both approaches lead to different results, not because of uncertainties in the method, but because they give answers to different questions. Here, we use the following wording to discriminate between these two types of questions and methods, knowing that other authors may use them differently: The impact of a source is calculated by the sensitivity method (here the perturbation approach), while the contribution is**

calculated using the source apportionment method (here tagging approach, e.g., Wang et al., 2009; Grewe et al., 2010; Clappier et al., 2017). Accordingly, the impact indicates the effect of an emissions change, while the contribution enables an attribution of ozone (and associated radiative forcing) to specific emissions sources.

*The IPCC AR5 WG1 Chapter 8 defined radiative forcing as "an instantaneous change in net (down minus up) radiative flux (shortwave plus longwave; in W $m^{-2}$) due to an imposed change." AR5s definition is generally consistent with previous definitions (e.g., Seinfeld and Pandis 2006; Jacob 1999). Contribution as defined as the combinatorial tagging is not consistent with an imposed change. First, there is no imposed change. In fact, removing those emissions (tra or shp) would not impose a change of similar magnitude. Thus, the idea that transport or shipping contributes to RF proportionally to combinatorial tagging is conceptually flawed.*

Reply: We are not sure, if we understand this comment correctly. From what we understand, referee#1 is arguing that only with the perturbation approach (e.g. by removing the traffic emissions) a radiative forcing (RF) could be calculated. If so, this is an important point and the referee's comment indicates that we need to clarify our RF calculations in more detail to show that it is actually largely in agreement with the IPCC RF definition. To clarify this, we start with the IPCC definition of the tropospheric ozone RF, which is the RF for the ozone change between 1850 and a current situation. We are here interested in attributing this RF to individual source of ozone, such as land transport emissions. For this, we need to know the ozone attributable to the respective emission source. If we add up all RFs for different emission sources based on ozone fields calculated by the perturbation approach, the sum of the RFs calculated for different emission sources is drastically lower than the total tropospheric ozone RF (e.g. Grewe et al., 2012). Hence, the use of the perturbation approach is not in line with the IPCC definition to attribute different emission sources to ozone (see also the simplified sketch in Fig. S1 which is also part of the revised Supplement).

In contrast, the idea of the tagging approach is to attribute the RF of $O_3$ proportional to the share of $O_3$ corresponding to the individual emission sources (as performed in a previous study by Dahlmann et al., 2011). The benefit of using the contribution of an emission source (in contrast to using the impact of the emission source) is that for the contribution the sum of the individual radiative forcings is equal to the total RF, i.e. $\sum_i^n RF^i \approx RF$ with $RF^i$ being the radiative forcings of the individual emission source $i$ of $n$ total emission sources. This does not hold for the perturbation approach (Grewe et al., 2012). To add more details of our approach, we moved the description of the RF calculations from Sect. 6 to Sect. 2 and added further explanations. In addition, we added some details concerning the assumptions used in this method in the Supplement. The description of our RF method in Sect. 2.3 is now:

**The radiative forcing (RF) of ozone is defined as the net flux change caused by a change (e.g. between two time periods like pre-industrial and present day, Myhre et al., 2013). Here, we are interested in the**

contribution of land transport and shipping to this RF. Due to the non-linearities in the ozone chemistry (see also Sect. 4), we estimate the contribution of the land transport/shipping emissions to ozone and then calculate the RF of these $O_3$ shares individually. This approach is consistent with the IPCC RF definition, since the sum of all individual RF contributions approximately equals the total RF (for a detailed example see Dahlmann et al., 2011).

Thus, to calculate the $O_3$ RFs of land traffic and shipping emissions, additional simulations were performed applying the stratospheric adjusted radiative forcing concept (e.g. Hansen et al., 1997; Stuber et al., 2001; Dietmüller et al., 2016). For this, monthly mean fields of the simulation *RC1SD-base-10a* are used as input data, of the radiation scheme, except for $O_3$, which stem from the *BASE* simulation. Calculations of the RF based on the results of the tagging approach in accordance with Dahlmann et al. (2011) were performed as follows:

1. Based on the results of the *BASE* simulation, monthly mean values of $\Delta_T^{\mathrm{tra}}$=$O_3$ - $O_3^{\mathrm{tra}}$ and $\Delta_T^{\mathrm{shp}}$=$O_3$ - $O_3^{\mathrm{shp}}$ were calculated. $\Delta_T^{\mathrm{tra}}$ and $\Delta_T^{\mathrm{shp}}$ corresponds to the share of $O_3$ excluding $O_3$ from land transport and shipping emissions, respectively.

2. Multiple radiation calculations (Dietmüller et al., 2016) were performed, calculating the radiative flux of $\Delta_T^{\mathrm{tra}}$, $\Delta_T^{\mathrm{shp}}$ and $O_3$. The $O_3$ RFs of land transport and shipping emissions using the tagging approach are then calculated as follows:

$$\mathrm{RF}_{\mathrm{O3tra}}^{\mathrm{tagging}} = rflux(O_3) - rflux(\Delta_T^{\mathrm{tra}}), \tag{1}$$

$$\mathrm{RF}_{\mathrm{O3shp}}^{\mathrm{tagging}} = rflux(O_3) - rflux(\Delta_T^{\mathrm{shp}}), \tag{2}$$

with $rflux$ being the radiative fluxes calculated for the respective quantity. Accordingly, the calculated RFs measure the flux change caused by the ozone share of land transport and shipping emissions, respectively.

Calculating the RFs based on the results of the perturbation approach is similar to (e.g. Myhre et al., 2011). First, $\Delta O_{3\mathrm{tra}}$ and $\Delta O_{3\mathrm{shp}}$ are calculated by taking the difference between the unperturbed (*BASE*, see below) and the perturbed simulations (*LTRA95* or *SHIP95*):

$$\Delta O_3 = (O_3^{\mathbf{unperturbed}} - O_3^{\mathbf{perturbed}}) \cdot 20. \tag{3}$$

As we consider 5 % perturbations these differences are scaled by a factor of 20 to yield a 100 % perturbation. To calculate the RFs using the perturbation approach, $\Delta O_{3\mathrm{tra}}$ and $\Delta O_{3\mathrm{shp}}$ are than treated as described above for $\Delta_T^{\mathrm{tra}}$ and $\Delta_T^{\mathrm{shp}}$. These RFs are called $\mathrm{RF}_{\Delta\mathrm{O3tra}}^{\mathrm{perturbation}}$

and $\text{RF}^{\text{perturbation}}_{\Delta \text{O3shp}}$, respectively. Accordingly, the method to calculate the RFs of the $O_3$ shares analysed by the perturbation and the tagging approach are the same. The differences between $\text{RF}^{\text{perturbation}}_{\text{O3tra}}$ and $\text{RF}^{\text{tagging}}_{\text{O3tra}}$ (and the same for shipping) arise only due to differences of the the differently calculated $O_3$ shares.

The benefit of using the contribution of an emission source (in contrast to using the impact of the emission source) is that for the contribution the sum of the individual radiative forcings is equal to the total RF, i.e. $\sum_i^n RF^i \approx RF$ with $RF^i$ being the radiative forcings of the individual categories $i$ of $n$ total categories. This hold for the perturbation approach (Dahlmann et al., 2011; Grewe et al., 2012). However, the calculations of the RF is still subject to some specific assumptions, which we discuss in detail in the Supplement.

*The authors assert that this technique is useful in understanding changes in emissions (particularly section 4.1). The current state of practice uses an emission reduction matrix to explore sensitivities at multiple emissions reductions (20, 40, 60, 80%) of both NOx and VOC. How is tagging this technique more useful than the iterative NOx/VOC matrix?*

Reply: We think that there is a misunderstanding. In the conclusion (last sentences) we clearly state:

'To investigate mitigation options, the tagging method cannot replace sensitivity studies and vice versa. However, we clearly demonstrated that a combination of both methods strengthen the investigation of mitigation options and should be the method of choice.'

As demonstrated in Sect. 4.1 we prefer to apply the tagging method in all sensitivity simulations performed at different emission reduction levels. This is important, because in a non-linear system the success of a particular mitigation option (e.g. reducing road traffic emissions by 10 %) strongly depends on the history of previous emission reductions. For instance in this case the sensitivity method measures the success of all mitigation options, while the additionally applied tagging method provides a more in depth understanding. The additional tagging method helps in attributing the remaining ozone to different sources and demonstrates that, for instance, emissions from industry contribute more to ozone after land transport emissions are reduced, because the ozone production efficiency of the industry emissions increase.

As discussed in the answer to referee#2 we rephrased Sect. 4.1 (page 13–14 of the revised manuscript) to make this more clear. In Addition, we changed the sentence above to: **To investigate mitigation options, the tagging method cannot replace sensitivity (i.e. perturbation) studies and vice versa.**

*Finally, I have concern about the methodology as described in Eq 2. Apportionment based on fraction of NOy and NMHC concerns me. See Page,Line*

*comments.*
Reply: Please see below for a detailed answer.

*Much of this critique is specific to the interpretation and assertions of unique value. The methods and results are internally consistent. I am skeptical of the species family approach as described. The biggest issue is that the article attempts to fully own the term "contribution", applies combinatorial tagging to RF in an odd way that needs to be clearly distinguished from traditional RF, and implies regulatory value that is likely already met. Most of these comments can be addressed by revising the interpretation.*
Reply: As described in detail (above and below) we changed parts of the manuscript to clarify the differentiation between impact and contribution.

*1,3: recommend "complementary" because the dynamics of "competition"*
Reply: Both, VOC and $NO_x$, are precursors of ozone and both species are attributed to ozone in our approach, as well as in the approaches by (e.g. Dunker et al., 2002; Kwok et al., 2015). In this sense $NO_x$ and VOC compete for the production.
Since the wording seems to confuse, we have rephrased the sentence:
**...but also their non-linear interaction in producing ozone.**

*1,5-7: The regions are not clear in the abstract. Consider adding "ocean" to each region to be consistent with text and clarify.*
Reply: Thanks! Added in the abstract and the conclusion!

*1,20: This is a narrow definition of the word contribution and I have seen no argument that combinatorial tagging is the only way to define contribution.*
Reply: As discussed above, we not to intend to restrict tagging only to our combinatorial approach, but to all tagged tracer approaches. We added 'source apportionment' in the first paragraph of the abstract to make this more clear. In the Introduction we also added some more details (see above):
**We quantify the contribution of land transport and shipping emissions to tropospheric ozone for the first time with a chemistry-climate model including an advanced tagging method (also known as source apportionment), which considers not only the emissions of $NO_x$ (NO and $NO_2$), CO or volatile organic compounds (VOC) separately, but also their non-linear interaction in producing ozone.**

*2,15: It is not important to know "contribution" as defined by combinatorial tagging to define mitigation strategies. In fact, knowing sensitivity is fundamentally more important to mitigation since the mitigation intends to impose a change.*
Reply: Indeed sensitivities are important to measure mitigation options, but it is also important to know which emission source contributes most to the ozone budget, in order to investigate, which emission sectors are worth to mitigate.

We rephrased the introduction to make this more clear (see above).

*3,4: F should be f*
Reply: Thanks! Changed!

*4,23-5,2: If implemented as discussed, this approach assumes two things that are fundamentally at odds with our understanding of atmospheric chemistry. First, it assumes that all NOy (NOx + NOz) is equally available for ozone production. This is problematic because NOy photochemical lifetime is much longer than NOx. As a result, this Eq 2 will attribute ozone production to NOx and NOz proportionally. That would lead to ozone being attributed to* $HNO3^{t}ag$ *in the mid to upper troposphere. Unless NOy is being defined differently than the field convention, this is troubling. Second, and less concerning, NMHC are not all equally reactive nor do they have equal RO2 yields. Assuming concentration fractions are proportional to combinatorial contribution is not consistent with the chemical mechanism.*
Reply: As discussed in Sect. 7, we are aware of the simplifications of the family approach. These simplifications are necessary in order to have a reasonable balance between complexity of the model and the demand regarding the computational resources (see discussion by Grewe et al., 2017). However, it is important to keep in mind that our tagging method relies on the diagnosed production and loss rates from the chemical solver (MECCA, Sander et al., 2011). MECCA calculates the $O_3$ production rates for each member of the $NO_y$ family individually, according to their kinetic rate coefficient (e.g. no $O_3$ is produced in regions, where only $HNO_3$ is present, see also our chemical mechanism in the Supplement). The family concept in the tagging method, however, can under certain circumstances indeed lead to a missatribution of ozone. Consider a case in which $O_3$ is locally produced from lightning $NO_x$ emissions. Using the family approach the tagged $NO_y$ family locally may consists also of $HNO_3$ from e.g. anthropogenic emissions. Accordingly, some of the produced $O_3$ would be attributed to anthropogenic emissions instead of the lightning emissions. This effect has been investigated by Grewe (2004), who concludes that this effect is important mainly during the first 12 h after a major emission and during this time may lead to an error caused by the family concept of up to 10 %.
Wee added a note on this in Sect. 7:
**Grewe (2004) showed for a simple box model that the implementation of the $NO_y$ family causes an error mainly after the first 12 h after major emission and during this time may lead to and error caused by the family concept of up to 10 %.**

*5,23: Februar[y]*
Reply: Thanks! Fixed!

*6,12: Is the seasonality of non-traffic reasonable and expected?*
Reply: Yes. The sectors 'Energy' and 'Residential' are important contributors to the non-traffic emissions, especially during winter, e.g. due to heating. For a

comparison, Fig S2 shows the monthly total anthropogenic non-traffic emissions of the MACCity (used in our study) and the EDGAR emission inventory.

*6,24: Why is July most comparable? What did those studies look at?*
Reply: In all other studies $O_3$ impacts for July conditions are presented. Therefore, we report our values also for July conditions. We changed the sentence to make this more clear:
**Please note that we list our values in Table 3 for July conditions only, to be comparable to other studies, since they also reported values for July conditions.**

*7,3: Reword or edit grammar*
The sentence "However, compared to other 5 % studies our results show, especially for NA, slightly larger values. This might be caused by a different geographical distribution and larger CO and NMHC emissions in our applied emission inventory. " was changed to:
**However, in general our simulation results show larger values compared to these previous findings. These differences are noticeable especially for the NA region. The differences might be caused by a different geographical distribution of the emissions, as well by larger CO and NMHC emissions in the emission inventory we applied.**

*7,8: This assumes that contribution == tagging, which the authors need to further consider.*
Reply: As discussed above the differentiation between perturbation (impact) and tagging (contribution) is well known and discussed in more detail in the references provided in this sentence. We rephrased this sentence and add a new reference (Clappier et al., 2017) to make this more clear:
**The comparison of our results using the 5 % perturbation approach with the results using the tagging approach clearly confirms the known differences between estimates of the impact (perturbation) and contribution (tagging) (e.g. Wang et al., 2009; Grewe et al., 2010; Emmons et al., 2012; Grewe et al., 2012, 2017; Clappier et al., 2017).**

*9,1-4: Are these ratios of partial column or average ratios?*
Reply: We always consider partial columns up to 850 hPa in DU in Sect. 4. We rephrased the paragraph slightly to make this more clear. In addition we added a proper unit to Fig.4. The sentence is now:
**To investigate this effect in more detail, $\Delta O_{3\text{tra}}$ (see Eq. 3) is analysed further. Here, we consider not only ground-level values, but partial ozone columns integrated from the surface up to 850 hPa (called 850PC, in DU).**

*10,2: consider replacing "almost" with "closest to".*
Reply: Thanks! Changed!

*10,7-12: Are not mitigation strategies more aligned with sensitivities?*
Reply: Of course, the success of a mitigation strategy is measured for instance by the reduction of ozone. This can be assessed with the perturbation approach. However, the perturbation approach does not give any information about changes of the ozone production efficiency from one sector, if other emissions are changed. This can be achieved with the tagging approach. Therefore, we propose to combine both methods (see next answer).

*10,20-31: See discussion of sensitivity matrix, which is the current approach for developing mitigation.*
Reply: As noted above we do not propose to replace sensitivity studies with tagging simulations, because tagging cannot replace perturbation to investigate the successes of a mitigation strategy. We propose to combine both methods, because the success of a mitigation measure depends on the sensitivity. Therefore, the success of one individual emission reduction strongly depends on the history of all previous emission reductions. The perturbation approach provides the general 'sucess' with respect to changes in ozone, while the results of the tagging approach allow an in-depth understanding of the results, an attribution of ozone to emission sources, and show how the production efficiency of other emission sources increase, if for instance road traffic emissions are decreased.
We largely rephrased Sect. 4.1 (see page 13–14 of the revised manuscript) making this point more clear:

**The tagging approach does not give any information about the sensitivity of the ozone chemistry with respect to a change of emissions. ....**
**A combination of tagging and perturbation is a powerful tool for putting additional pressure on unmitigated emission sources, because, even if the absolute ozone levels do not change, their shares in high ozone values (or radiative forcing) increase.**

*11,20: "[global] land transport." This section is tricky because the production may come from upwind sources. Try to be more explicit.*
Reply: This is indeed a very good point. To make this more clear we added an additional sentence at the beginning of the paragraph:
**Please note, in our tagging method we distinguish only between different emission sources, but not between emission regions. Therefore, the budgets analysed for distinct geographical regions might not be solely influenced by regional emissions, but also by upwind sources.**

*13,24: Be more specific than 'some'.*
Reply: We changed the sentence accordingly:
**Recent updates of the tagging scheme with respect to differences of the $HO_x$ family show an influence of 1–3 percentage-points on the relative contribution of land transport and shipping emissions (Rieger et al., 2017).**

*13,25: trough −> through?*
Reply: Yes! Tanks!

*13,25: is the author referring to engineering simplifications in the CTM?*
Reply: Yes. We rephrased the sentence:
**Therefore, we conclude that the error through the simplifications of the tagging method is estimated to be smaller than the errors arising from approximations applied in the global chemistry-climate-models itself (physics and chemistry parameterisations, e.g. 20 % given by Eyring et al., 2007).**

*13,28-29: CAMx OSAT/APCA[camx.com] and CMAQ ISAM [doi: 10.5194/gmd-8-99-2015] are a couple of examples of similar complexity to this scheme.*
Reply: We are well aware of these approaches, which are mainly used in regional air quality modell (and, to our knowledge are not used in global chemistry-climate models). However, as discussed, these approaches are based on thresholds of the $NO_x$/VOC sensitivity, well chosen for the intended purpose. But they are not comparable to our approach, which accounts for the competing effects between all species. Approaches by Emmons et al. (2012) or Butler et al. (2011) are also available on the global scale, but consider either only NOx or VOC only. We rephrased this sentence:
**Other available tagging schemes, however, are based on kinetic approaches (Gromov et al., 2010), consider either only $NO_x$ or VOC (e.g. Emmons et al., 2012; Butler et al., 2011), or are based on thresholds depending on whether the ozone chemistry is $NO_x$ or VOC limited (e.g. Dunker et al., 2002; Kwok et al., 2015). The differences between the assumptions and the scales on which they are applied render a detailed comparison impossible.**

*14,22-24: One interpretation is that the radiative forcing in this paper is an overestimate due to the lack of realism in the tagging compared to an actual imposed response.*
Reply: We do not agree with Referee#1 on this point. The larger RFs using the tagging approach compared to the perturbation approach are due to larger ozone shares. As discussed above, the methodology of calculating the RFs is the same between tagging and perturbation. However, to make this more clear we add zonal averages of the contribution and the impact of both emission sources to the Supplement and to this reply (see Fig. S3). Further, we stressed this point in more detail in Sect. 6 and in the conclusion. In Sect.6 the following note were added:

[revised manuscript text omitted]

Figure S1: Simplified sketch of three different ways to calculate RFs. 'RF O3' shows the classical way of calculating the anthropogenic RF by calculating the radiative flux of an preindustrial simulation and a simulation with all emissions. 'Perturbation' shows the perturbation approach, here the RF of different emission sources is estimated by perturbation simulations turning specific emissions off. This approach, however, leads to a part of ozone which can not be attributed to one sector (marked with ?). This is mainly caused by changes of the ozone production efficiency. The 'tagging' method estimates a radiative forcing for every specific category. Accordingly, a complete attribution of the RF to specific emission sources is possible.

[Figure]

Figure S2: Globally integrated $NO_x$ emissions (in Tg (NO) per month) of the anthropogenic non-traffic sector for the MACCIty emission inventory (red) and the EDGAR 4.3.1 inventory (black). Shown are values for the year 2010 exemplarily.

[Figure]

Figure S3: Multi-annual zonal average (2006–2010) $O_3$ shares as estimated by the perturbation method and the tagging approach. Shown are the contribution and impact of the land transport and shipping emissions to ozone, as estimated by the tagging method and the perturbation approach, respectively.

---

## Author Comment (AC2) · 16 Feb 2018

We thank referee#2 for many useful comments, which helped to improve the manuscript. In the following, referee comments are given in italics, our reply's in normal font, and text passages which we included in the text, in bold.

*This paper offers a nice overview of the impact of shipping emissions on ozone through the use of two methodologies: the tagging methodology and the perturbation methodology. The paper is well written and extremely thorough with a clear comparison to previous studies.*

Reply: We thank referee#2 for this very positive and encouraging comments.

*1. The authors state two goals in this study (p3, l4-6) in determining ozone from shipping emissions: to review previous studies and to give the results of the tagging method. The results from the authors use of the tagging methodology nicely complements estimates from the contribution method. I think the paper works as a review paper. However, as written, I question whether the paper stands very well on its own as a new piece of research. There does not seem to be enough new. Part of the author's justification for this paper is that no one has investigated the ozone contributions from transportation using the tagging approach. Just because something has not been done does not mean it is scientifically interesting or worth pursuing. There are probably other emission sectors that have not been investigated using the tagging approach. It doesn't seem that there should be a new paper written for each of these sectors. I think the authors need to better justify their study than simply state it has not been done. Why do we need another paper on the emission contributions from transportation emissions given the uncertainty? Specifically, what new insights does the tagging approach give? (This needs to be better clarified, see below). What do we learn about the tagging approach here that we didn't know before?*

Reply: We thank Referee#2 for acknowledging the review character of our manuscript. Of course, the general difference between tagging and perturbation is well known and has been discussed in many studies (which we cite in the manuscript), especially for simplified models. Of course, it might not be worth to study the difference between impact and contribution for each sector in detail. However, land transport and shipping emissions are very important anthropogenic emission sectors and are therefore subject to mitigation. Further, we want to highlight the following points:

- We here confirm previous results (using different methods) with a new method. This is very important, in particular this shows that those results are robust. Moreover, reproduciblability with different methods is an important aspects in science.

- To our knowledge, we applied for the first time the tagging and the perturbation approach simultaneously and consistently for land transport and shipping emissions,

- including a consistent way of calculating the radiative forcing (RF), thus allowing for a detailed comparison of the results.

- Further, we consider for the first time in a chemistry-climate-model the interactions between $NO_x$ and VOC. Our results indicate that the RFs calculated by Dahlmann et al. (2011) and Grewe et al. (2012) using a $NO_x$ only tagging are likely too large. Accordingly, we present new best estimates of the ozone RF, which are between previous estimates using the perturbation and the $NO_x$ only tagging.

- In addition, the tagging method allows us to present detailed results with respect of the influence of the land transport and shipping emissions on the tropospheric ozone budget.

To stress these aspects more, we revised the Introduction, Section 4/6 and the Conclusion. Please find the detailed differences in the 'diff version' of the revised manuscript.

*2. Why does the present study use a 5% perturbation? The results are sensitive to this perturbation. Some justification is needed. It would be helpful for comparison purposes if the authors also gave their results for a 100% perturbation in their tables. To what extent does the discrepancy with the tagging method come from the assumed 5% reduction? It appears a 100% emission reduction gives similar results to the tagging method. Reporting on a 100% emission perturbation would also help compare with other studies.*
Reply: As discussed in previous studies, the small perturbation approach minimises the impact of non-linearity. A 100% perturbation is considered as not being realistic (e.g. Hoor et al., 2009; Grewe et al., 2010; Koffi et al., 2010).
In the revised manuscript we added a note on this in Sect. 4:
**The 5 % perturbation was chosen as previous studies showed that this small perturbation sufficiently minimises the impact of the non-linearity of the chemistry on the results (e.g. Hoor et al., 2009; Grewe et al., 2010; Koffi et al., 2010).**

*3. Equation (3): Is a factor of 20 missing?*
Reply: In the first version of our manuscript we focused on differences between the 5 % perturbations. Accordingly, no factor was missing. In the revised manuscript we revised this part of the manuscript (see below) and made the factor of 20 more clear.

*4. The definition of gamma needs to be clarified in more detail in the text. After looking in detail at the figure and reading the text the meaning of gamma became clear, but it should not have been this difficult. Please clarify the definition of gamma in the text explicitly stating what the y intercept is and stating that y is the average net ozone production rate in a particular region.*
Reply: We rephrased the section about $\Gamma$ considering also your next point to make the definition of $\Gamma$ more clear. Our definition of $\Gamma$ is also used in science

of economics. There, elasticity ($\eta$) is defined as $\eta = 1 - \Gamma$. In economics $\eta$ measures the change of an economic variable, if another variable is changed. The changed paragraph is now:

**Based on the results of the *REF* and *LTRA95* simulations, the ozone sensitivity is calculated with the tangent approach in accordance with Grewe et al. (2010) by solving a linear equation ($y = m \cdot (x - x_0) + b$). Here, $x$ and $y$ are the average $NO_x$ mixing ratio and the net $O_3$ production ($P_{O3}$), respectively, for a particular region. The $m$ denotes the slope, which corresponds to an approximation of the derivative $dP_{O3}/dNO_x$ in the unperturbed simulation, which is calculated by the difference in ozone production and $NO_x$ mixing ratios in the unperturbed and perturbed simulation. $x_0 = NO_x^u$ is the $NO_x$ mean mixing ratio in the unperturbed simulation and $b = P_{O3}^u - dP_{O3}/dNO_x\ NO_x^u$, where $P_{O3}^u$ is the mean ozone production in the unperturbed simulation.**

**Based on the linearised ozone production ($P_{O3}^{lin}$) calculated by the tangent approach, we define a saturation indicator $\Gamma$, which helps to analyse the ozone sensitivity further:**

$$\Gamma = \frac{\mathrm{y-axis\ intercept}}{\mathrm{y-value\ of\ unperturbed\ simulation}} = \frac{P_{O3}^{lin}(NO_x = 0)}{P_{O3}^{lin}(NO_x = \mathrm{unperturbed})}. \quad (1)$$

**This value is a quantitative indicator of the chemical regime, showing how much an emission change of one specific sector is compensated by increased ozone productivity of other sectors. $\Gamma = 1$ indicates a saturated behaviour of the ozone production i.e. the ozone production does not change, if emissions are changed ($P_{O3}^{lin}(NO_x = 0) = P_{O3}^{lin}(NO_x = \mathrm{unperturbed})$). Accordingly, there is no ozone reduction because the change of the emissions is entirely compensated by the increase of the ozone production efficiency of other emissions. $\Gamma > 1$ indicates an overcompensating effect, i.e., reduced $NO_x$ emissions lead to an increase of the ozone production (corresponding to the VOC-limited regime). Finally, $\Gamma = 0$ indicates a linear response of the system (with a y-intercept at zero). Accordingly, the ozone change introduced by an emission change is not compensated by an increase of the ozone production efficiency. For $\Gamma = 0.5$ the ozone change is half compensated by a change in the ozone production efficiency. In terms of the estimated derivative ($dP_{O3}/dNO_x$), $\Gamma = 1$ corresponds to $dP_{O3}/dNO_x = 0$, while $\Gamma > 1$ corresponds to $dP_{O3}/dNO_x < 0$ and vice versa.**

*5. It is unclear why gamma is defined in terms of the intercept instead of the slope (dO3/dNOx). The intercept will be leveraged by the amount of the NOx emissions. That is, the impact of the slope the will be amplified when the NOx emissions are large by changing the intercept to a greater extent than if the NOx emissions are small.*

Reply: Of course $\Gamma$ could also be defined in terms of the slope (dPO3/dNOx). However, we use $\Gamma$ as indicator to check whether the ozone production increases, decreases or stays the same with changed emissions. Exactly the same were possible using the slope. To make this more clear we added a comparison between the slope and $\Gamma$ for the different regimes (see above).

*6. Figure 5 clearly demonstrates that the perturbation approach gives different estimates under different conditions. However, it does not show how the tagging approach differs. Some more work is needed here to better understand how these two approaches give different answers depending on ambient conditions and transportation emissions. From line 9 onwards (on page 9) the well-known dependence of ozone production on NOx is shown, with the well-known result that in regions of high NOx a decrease in emissions has little impact on the ozone concentration. There is not much new here. The text and figures don't explicitly show that the tagging approach gives a different answer than the perturbation approach. And isn't the discrepancy between the two methods well known. What is new?*

Reply: New is the quantification of the competing effects by combining tagging with the perturbation method and the calculation of the $\Gamma$ value. Of course, the response of the ozone chemistry to $NO_x$ emissions, as well as the difference between impact and contribution, are well known. We clearly state this in our text and refer to previous publications. It shows that the basic chemical response is in line with previous studies, forming the base for a better understanding and quantification of the underlying processes. We revised the Section 4 (including 4.1 see below) in large parts to quantify the difference between tagging and perturbation in more detail. Please see page 11–14 of the revised manuscript for the changed sections:

**As discussed in the previous section and by previous studies (e.g. Wang et al., 2009; Grewe et al., 2010) the perturbation approach**

**.....**

**even if the absolute ozone levels do not change, their shares in high ozone values (or radiative forcing) increase.**

*7. The authors state: 'Combining the tagging and the perturbation approach is there- fore the best way to measure the success of a mitigation strategy.' (p10, l19-20). The authors argue that the perturbation approach gives different answers depending on the current state. I suppose the tagging approach gives the same ozone reduction regardless of the mitigation pathway. This should be clearly stated. Nevertheless, it is unclear how one would use the tagging method to decide on mitigation issues. Perhaps a concrete example would be helpful here? This is important because it would provide a needed justification of the tagging approach. It is crucial that the paper clearly gets this across. It seems to me the tagging approach is useful in assigning blame: for example, if you want to apportion blame for an ozone pollution outbreak or for the radiative forcing due to ozone. It is not clear to me how one would use the tagging method practically in assessing mitigation options.*

Reply: We are very thankful to referee#2 for this comment. Of course it is very important to us to get the benefit of combining tagging and perturbation across. Obviously in the first manuscript this point was not stressed enough. Therefore, we revised Subsection 4.1 in large parts (.

*8. The loss rate of ozone is very dependent on how it is calculated (page 11). How are the losses calculated in the present study? Are they calculated in the same manner in the comparison studies?*
Reply: We considered the following loss rates (cf. equation 14 in Grewe et al. (2017)):

- reactions of $O_3$ with OH and $HO_2$,

- effective loss reactions of $O_3$ with $NO_y$ species,

- reactions of $O_3$ with NMHCs, and

- reactions mainly of $O^1(D)$ with different species (e.g. $O^1(D)$ + $H_2O$) leading to an effective $O_3$ loss.

We added our detailed chemical mechanisms which indicates the reactions, which are considered for effective loss and production of $O_3$ to the Supplement. We added a note on this in our description of the tagging method:
**The chemical mechanism including all diagnosed production and loss rates for the tagging method are part of the Supplement. The analysed production and loss rates in Sect. 5 are calculated in accordance with Eq. 13 and 14 of Grewe et al. (2017).**

Indeed the values presented by Young et al. (2013), which we use for comparison, are results of a multi-model intercomparison. As stated by Young et al. (2013) not all models, which participated in the intercomparision, calculate ozone loss in a comparable manner (exact details, however, are not given). We added a note on this in the revised manuscript:
**Further, it is important to note that loss rates are not calculated consistently in all models presented by Young et al. (2013).**

*9. P12, l 16: 'We obtain. . ..' Using which method?*
Reply: To make this more clear we differentiate in the revised manuscript between $RF_{O3tra}^{tagging}$ and $RF_{\Delta O3tra}^{perturbation}$ (which we define in Sect. 2):
**We obtain a global net RF for land transport of $RF_{O3tra}^{tagging} = 92$ mW m$^{-2}$. The shortwave RF is 32 mW m$^{-2}$ and the longwave RF is 61 mW m$^{-2}$. The estimated RF of ship traffic is $RF_{O3shp}^{tagging} = 62$ mW m$^{-2}$ and smaller than the land transport RF.**

*10. The section on uncertainties should also discuss the uncertainties in the perturbation method. In particular this method is sensitive to the perturbation assumed.*

Reply: Thanks. This is indeed a good point. We added:

**However, also the perturbation approach faces an important limitation. The calculated impact largely depends on the magnitude of the chosen perturbation and the impacts are only valid for this specific perturbation (e.g. Hoor et al., 2009). In addition, the perturbation approach has a fundamental problem, namely a non-closed budget. This means that the sum of $O_3$ changes calculated for different perturbed emission sources (e.g. land transport and aviation) is not necessarily the total $O_3$ change if all emissions are reduced at the same time (e.g. Wang et al., 2009; Grewe et al., 2010).**

**References**

Dahlmann, K., Grewe, V., Ponater, M., and Matthes, S.: Quantifying the contributions of individual NOx sources to the trend in ozone radiative forcing, Atmos. Environ., 45, 2860–2868, doi:http://dx.doi.org/10.1016/j.atmosenv.2011.02.071, URL http://www.sciencedirect.com/science/article/pii/S1352231011002366, 2011.

Grewe, V., Tsati, E., and Hoor, P.: On the attribution of contributions of atmospheric trace gases to emissions in atmospheric model applications, Geosci. Model Dev., 3, 487–499, doi:10.5194/gmd-3-487-2010, URL http://www.geosci-model-dev.net/3/487/2010/, 2010.

Grewe, V., Dahlmann, K., Matthes, S., and Steinbrecht, W.: Attributing ozone to NOx emissions: Implications for climate mitigation measures, Atmos. Environ., 59, 102–107, doi:10.1016/j.atmosenv.2012.05.002, URL http://www.sciencedirect.com/science/article/pii/S1352231012004335, 2012.

Grewe, V., Tsati, E., Mertens, M., Frömming, C., and Jöckel, P.: Contribution of emissions to concentrations: the TAGGING 1.0 submodel based on the Modular Earth Submodel System (MESSy 2.52), Geoscientific Model Development, 10, 2615–2633, doi:10.5194/gmd-10-2615-2017, URL https://www.geosci-model-dev.net/10/2615/2017/, 2017.

Hoor, P., Borken-Kleefeld, J., Caro, D., Dessens, O., Endresen, O., Gauss, M., Grewe, V., Hauglustaine, D., Isaksen, I. S. A., Jöckel, P., Lelieveld, J., Myhre, G., Meijer, E., Olivie, D., Prather, M., Schnadt Poberaj, C., Shine, K. P., Staehelin, J., Tang, Q., van Aardenne, J., van Velthoven, P., and Sausen, R.: The impact of traffic emissions on atmospheric ozone and OH: results from QUANTIFY, Atmos. Chem. Phys., 9, 3113–3136, doi:10.5194/acp-9-3113-2009, URL http://www.atmos-chem-phys.net/9/3113/2009/, 2009.

Koffi, B., Szopa, S., Cozic, A., Hauglustaine, D., and van Velthoven, P.: Present and future impact of aircraft, road traffic and shipping emissions on global tropospheric ozone, Atmos. Chem. Phys., 10, 11 681–11 705, doi:10.5194/acp-10-11681-2010, URL `http://www.atmos-chem-phys.net/10/11681/2010/`, 2010.

Wang, Z. S., Chien, C.-J., and Tonnesen, G. S.: Development of a tagged species source apportionment algorithm to characterize three-dimensional transport and transformation of precursors and secondary pollutants, Journal of Geophysical Research: Atmospheres, 114, n/a–n/a, doi:10.1029/2008JD010846, URL `http://dx.doi.org/10.1029/2008JD010846`, d21206, 2009.

Young, P. J., Archibald, A. T., Bowman, K. W., Lamarque, J.-F., Naik, V., Stevenson, D. S., Tilmes, S., Voulgarakis, A., Wild, O., Bergmann, D., Cameron-Smith, P., Cionni, I., Collins, W. J., Dalsøren, S. B., Doherty, R. M., Eyring, V., Faluvegi, G., Horowitz, L. W., Josse, B., Lee, Y. H., MacKenzie, I. A., Nagashima, T., Plummer, D. A., Righi, M., Rumbold, S. T., Skeie, R. B., Shindell, D. T., Strode, S. A., Sudo, K., Szopa, S., and Zeng, G.: Pre-industrial to end 21st century projections of tropospheric ozone from the Atmospheric Chemistry and Climate Model Intercomparison Project (AC-CMIP), Atmos. Chem. Phys., 13, 2063–2090, doi:10.5194/acp-13-2063-2013, URL `http://www.atmos-chem-phys.net/13/2063/2013/`, 2013.

---

## Author Response (AR2)

Dear editor,

thank you very much for further processing the editorial process.

According to the comments we revised Section 7 including a further discussion of the errors caused by the NOy- and NMHC- family approach.

Attached are the comments to the two referees (original comments in italic, answers in normal fonts, changes in the manuscript in bold) together with the revised manuscript. In the revised manuscript all modifications are highlighted (latexdiff).

We are looking forward to your reply,

Mariano Mertens (on behalf of all co-authors) We thank referee#1 for the useful comment. In the following, referee comments are given in italics, our reply's in normal font, and text passages which we included in the text, in bold.

The authors have responded well to the reviewers' comments and recommendations. I recommend two final comments. The first should be quite easy. The second I offer for consideration.

Reply: Thanks for your positive comment. Please see below for the detailed answers.

The authors have added a good description of Grewe (2004) to justify NOy,i/NOyNOx,i/NOx, but have not offered any such evidence for NMHC. The authors rightly point out that the MESSy solver diagnosed rates may implicitly account for much. That being said, the authors should acknowledge that the error associated with the NMHC has not been quantified. If it has been quantified, the authors should point the readers to it.

Reply: That is indeed correct. Up to now we can not quantify the error caused by the family approach for the NMHC family. With the current implementation of the tagging method this detailed quantification is not possible. However, we are working on more detailed error diagnostics. In the future, additional comparisons with new detailed VOC-tagging schemes (e.g. Butler et al., 2018) certainly help to quantify the error in more detail.

We added the following note in the revised manuscript:

Grewe (2004) showed that the implementation of the  $NO_y$  family causes an error mainly after the first 12 h after major emission and during this time may lead to an error caused by the family concept of up to 10 %. However, the analyses by Grewe (2004) have only been performed with a simple box model for the upper troposphere and considered only the  $NO_y$  family. Applied in an chemistry-climate model this error might be larger, especially with respect to the interplay of freshly emitted lightning- $NO_x$  emissions and oxidized anthropogenic emissions in the upper troposphere. A detailed quantification of this error is difficult. The implementation of the NMHC family causes an additional error, as the different reactivities are not explicitly taken into account. Currently this error cannot be quantified in detail. Other detailed VOC-tagging approaches might help to quantify this error (e.g. Butler et al., 2018).

The authors response to NOy/NMHC needs to be enhanced for radiative forcing with respect to shipping. Radiative forcing is particularly sensitive to ozone in the upper troposphere. Grewe (2004) showed that shipping contributed 1% (Fig 3) in the 300-100hPa range and that the Fa estimate was -1 to 1% (Fig 4). In this case, the difference for shipping in the free troposphere (where RF

is sensitive) is on the same order as the contribution itself. The sensitivity of all small contributors (e.g., shipping) can be inferred to be the direct result of unstable systems associated with Lightning NOx (Fa=-10.-20%) and strat intrusions. Given that Grewe (2004) attributes large amounts of ozone to lightning, some ozone production in the free troposphere likely occurs following a lightning injection - well within a few hours of the emission near the non-steady state period. Near the lightning NOx injection would be consistent with when Grewe (2004) showed highest sensitivity. However, the emissions in Grewe (2004) were continuous in both boxes. By contrast, lightning would be episodically fresh and shipping emissions would be consistently well oxidized by the time they reach the upper troposphere. The contrast between fresh lightning and oxidized shipping provides the exact conditions that this tagging methodology is sensitive to. The authors make a nod to this uncertainty by referencing Grewe (2004), but in this reviewers opinion the authors minimize the relevance. As the Grewe (2004) pointed out, the truth is elusive and the methodological comparison (Fa) does not necessarily identify a flaw in this tagging. However, the conceptual model of lightning in the upper troposphere provides a theoretical framework for questioning the tagged results in the region relevant to RF. I do not think this should stop the manuscript, but I also think the dismissal of the concern based on 10% from Grewe (2004) is overly simplistic.

Reply: Indeed, the 10 % error given by Grewe (2004) is only valid for the steady state case and derived with a simplified box model for the upper troposphere. We agree with the reviewer that in a 3D-model the error might be larger (especially for small emission sources). For the future, more detailed analyses of the error caused by the  $NO_y$  family approach, especially with respect to lightning- $NO_x$ , are necessary, given their large radiative efficiency Dahlmann et al. (2011) and the high sensitivity of ozone to RF in the upper troposphere.

In addition to the changes mentioned above we added a further note in Sect. 7:

Further, due to the large sensitivity of the RF to ozone in the upper troposphere in particular lightning- $NO_x$  shows a large radiative efficiency (Dahlmann et al., 2011) errors in the attribution due to the  $NO_y$  family approach (see above) can lead here to an overestimated RF. This needs to be investigated in more detail in the future.

pg 5,26 submodul should be submodule Reply: Thanks! It should be submodel. We changed this.

Thanks for the additional review and the valuable comment. In the following, referee comments are given in italics, our reply's in normal font, and text passages which we included in the text, in bold.

As the editor of this paper, I am providing a second review of this revised manuscript in order to keep the process moving forwards. I think the authors have done a good job in responding to the comments of the two anonymous reviews given in the open discussion phase. They have produced a much-improved manuscript. I agree with the one remaining comment on the revised manuscript from anonymous reviewer #1, that the error involved in treating NMVOC as a single ozone precursor family has not been adequately quantified or discussed. I agree with the suggestion by the anonymous reviewer, that the authors should address this remaining concern. I would consider it acceptable for the authors to mention this as an unquantified error somewhere in their Section 7, along with their discussion of the error due to the use of NOy as an ozone precursor family.

Reply: Thank you very much for honouring the work we put in the improved manuscript. As discussed in the answer to referee#1 we added a note about the simplification in Sect. 7 together with an improved discussion about the error caused by the simplifications of the  $NO_y$ -family in Sect. 7. The most important change is:

[revised manuscript text omitted]
_{R1}^{tag} = \frac{1}{2} P_{R1} \left( \frac{NO_y^{tag}}{NO_y} + \frac{NMHC^{tag}}{NMHC} \right).$$
(2)

In this case the variables marked with tag represent the tagged production rate of O3 by reaction R1 (PR1) as well as the tagged families of NOy and NMHC (details given below) of one individual category (e.g. land transport). Accordingly the fractional apportionment is inherent to the method based on a combinatorial approach, which decomposes every regarded reaction into all possible combinations of reacting tagged species. This takes into account the specific reaction rate constant from the full chemistry scheme (implicitly by the production and loss rates from the chemistry solver). The chemical mechanism
including all diagnosed production and loss rates for the tagging method are part of the Supplement. The analysed production and loss rates in Sect. 5 are calculated in accordance with Eq. 13 and 14 of Grewe et al. (2017).

The applied method considers ten categories (detailed definition is given in Table 1). To minimize the needed amount of memory and computational performance, not every individual specie is tagged. Instead a family concept is chosen. The following families are taking into account: O3, NOy, PAN, NMHC and CO. Additionally, OH and HO2 are tagged by a

steady state approach. In the following, we denote absolute contributions of land transport and shipping emissions to ozone diagnosed with the tagging method as  $O_3^{tra}$  and  $O_3^{shp}$ , respectively.

**2.3 Radiative forcing**

The radiative forcing (RF) of ozone is defined as the difference of the net radiative fluxes caused by a change (e.g. between two time periods like pre-industrial and present day, Myhre et al., 2013). Here, we are interested in the contribution of land transport and shipping to this RF. Due to the non-linearities in the ozone chemistry (see also Sect. 4), we estimate the contribution of

5 the land transport/shipping emissions to ozone and then calculate the RF of these  $O_3$  shares individually. This approach is consistent with the IPCC RF definition, since the sum of all individual RF contributions approximately equals the total RF (for a detailed example see Dahlmann et al., 2011).

Thus, to calculate the  $O_3$  RFs of land traffic and shipping emissions, additional simulations were performed applying the stratospheric adjusted radiative forcing concept (e.g. Hansen et al., 1997; Stuber et al., 2001; Dietmüller et al., 2016). For

- 10 this, monthly mean fields of the simulation *RC1SD-base-10a* are used as input data, of the radiation scheme, except for  $O_3$ , which stem from the *BASE* simulation. Calculations of the RF based on the results of the tagging approach in accordance with Dahlmann et al. (2011) were performed as follows:
  - 1. Based on the results of the *BASE* simulation, monthly mean values of  $\Delta_T^{\text{tra}}=O_3 O_3^{\text{shp}}=O_3 O_3^{\text{shp}}=O_3 O_3^{\text{shp}}$  were calculated.  $\Delta_T^{\text{tra}}$  and  $\Delta_T^{\text{shp}}$  corresponds to the share of  $O_3$  excluding  $O_3$  from land transport and shipping emissions, respectively.
  - 2. Multiple radiation calculations (Dietmüller et al., 2016) were performed, calculating the radiative flux of  $\Delta_T^{\text{tra}}$ ,  $\Delta_T^{\text{shp}}$  and O3. The O3 RFs of land transport and shipping emissions using the tagging approach are then calculated as follows:

$$RF_{O3tra}^{tagging} = rflux(O_3) - rflux(\Delta_T^{tra}),$$
(3)

$$RF_{O3shp}^{tagging} = rflux(O_3) - rflux(\Delta_T^{shp}),$$
(4)

20

15

[revised manuscript text omitted]
                     | 416                  | 15                    | 5                     |